Manuscript prepared for Atmos. Chem. Phys.
with version 2015/04/24 7.83 Copernicus papers of the LaTeX class copernicus.cls.
Date: 3 November 2016

# Assessing the sensitivity of the hydroxyl radical to model biases in composition and temperature using a single-column photochemical model for Lauder, New Zealand

Laura López-Comí[1,2], Olaf Morgenstern[1,*], Guang Zeng[1,*], Sarah L. Masters[2], Richard R. Querel[1], and Gerald E. Nedoluha[3]

[1]National Institute of Water and Atmospheric Research (NIWA), Lauder, New Zealand
[2]Department of Chemistry, University of Canterbury, Christchurch, New Zealand
[3]United States Naval Research Laboratory, Washington, DC, United States
[*]now at NIWA, Wellington, New Zealand

*Correspondence to:* O. Morgenstern (olaf.morgenstern@niwa.co.nz)

**Abstract.** We assess the major factors contributing to local biases in the hydroxyl radical (OH) as simulated by a global chemistry-climate model, using a single-column photochemical model (SCM) analysis. The SCM has been constructed to represent atmospheric chemistry at Lauder, New Zealand, which is representative of the background atmosphere of the Southern Hemisphere (SH) mid-latitudes. We use long-term observations of variables essential to tropospheric OH chemistry, i.e. ozone ($O_3$), water vapour ($H_2O$), methane ($CH_4$), carbon monoxide (CO), and temperature, and assess how using these measurements affect OH calculated in the SCM, relative to a reference simulation only using modelled fields. The analysis spans 1994 to 2010. Results show that OH responds approximately linearly to correcting biases in $O_3, H_2O, CO, CH_4$, and temperature. The biggest impact on OH is due to correcting an overestimation by approximately 20 to 60% of $H_2O$, using radiosonde observations. Correcting this moist bias leads to a reduction of OH by around 5 to 35%. This is followed by correcting predominantly overestimated $O_3$. In the troposphere, the model biases are mostly in the range of $-10$ to 30%. The impact of changing $O_3$ on OH is due to two pathways; the OH responses to both are of similar magnitude but different seasonality: Correcting in situ tropospheric ozone leads changes in OH in the range $-14$ to 4%, whereas correcting the photolysis rate of $O_3$ in accordance with overhead column ozone changes leads to increases of OH of $8-16\%$. The OH sensitivities to correcting $CH_4$, CO, and temperature biases are all minor effects. The work demonstrates the feasibility of quantitatively assessing OH sensitivity to biases in longer-lived species, which can help explain differences in simulated OH between global chemistry models and relative to observations. In addition to clear-sky simulations, we have performed idealised sensitivity simulations to assess the impact of clouds (ice and liquid) on OH. The results indicate that the impacts on the ozone photolysis rate and OH are substantial, with a general decrease of OH below the clouds of up to 30% relative to the clear-skies situation, and an increase of up to 15% above. Us-

ing the SCM simulation we calculate recent OH trends at Lauder. For the period of 1994 to 2010, all trends are insignificant, in agreement with previous studies. For example, the trend in total-column OH is $0.5 \pm 1.3\%$ over this period.

## 1 Introduction

The hydroxyl radical (OH) is essential to atmospheric chemistry as the leading oxidizing agent. It acts as a "detergent", reacting with numerous, mostly organic pollutants (Levy, 1971; Logan et al., 1981; Thompson, 1992; Lelieveld et al., 2004; Naik et al., 2013) and controls the lifetimes of many trace gases containing carbon-hydrogen bonds, particularly methane ($CH_4$), because reaction with OH is their dominant removal mechanism. It is also responsible for oxidizing atmospheric trace gases such as carbon monoxide (CO), non-methane volatile organic compounds (NMVOCs), and also some ozone-depleting substances such as hydrochlorofluorocarbons (HCFCs) (DeMore, 1996). Therefore, the oxidizing capacity of the atmosphere is largely defined by the abundance of OH. Tropospheric ozone ($O_3$), an air pollutant and greenhouse gas (GHG), is the primary source of OH in the troposphere. Although it only accounts for 10% of the total atmospheric $O_3$ abundance, it plays an essential role in photochemical processes controlling tropospheric composition. It forms OH via $O_3$ photolysis yielding excited oxygen ($O(^1D)$) and a subsequent reaction of $O(^1D)$ with water vapour ($H_2O$). $CH_4$ and CO oxidation by OH, and other oxidation processes involving NMVOCs, lead to formation of tropospheric $O_3$ in the presence of $NO_x$ (Logan et al., 1981; Thompson, 1992; Lelieveld and Dentener, 2000). In low-$NO_x$ atmospheric environments, such as in much of the SH, downward transport of $O_3$ from the stratosphere is the main source of tropospheric $O_3$, followed by $O_3$ transport from other regions where it is chemically produced (Zeng et al., 2010). Stratospheric $O_3$ also plays an important role through its impact on the $O_3$ photolysis rate $j_{O1D}$ which is affected by the overhead $O_3$ column. For instance, stratospheric $O_3$ depletion produces increased UV penetration to the troposphere. This affects the production of tropospheric OH.

The most widely used method for field measurements of OH is the Fluorescence Assay by Gas Expansion (FAGE) technique and is based on the measurement of OH and other species concentrations through ultra-violet (UV) laser induced fluorescence spectroscopy. OH measurements using the FAGE technique have been conducted in a large variety of atmospheric environments, ranging from polluted (Ren et al., 2003; Dusanter et al., 2009) to clean (Creasey et al., 2003; Bloss et al., 2007) atmospheres. However, due to its very short lifetime (the global lifetime is estimated to be $\sim 1$ s, Prinn, 2001; Elshorbany et al., 2012) and large variability, such in situ measurements of OH do not sufficiently capture its global abundance, which makes it difficult to sufficiently constrain global OH abundances with in situ measurements (Heard and Pilling, 2003). For that reason, modelling is an essential tool to study global OH. OH is routinely included in global models of tropospheric chemistry, but the complexity of the tropospheric chemical system and the sensitivity of

OH to a variety of environmental factors mean that there is considerable disagreement among global chemistry-transport and chemistry-climate models regarding the global OH abundance; this is often expressed in terms of the $CH_4$ lifetime (e.g., Stevenson et al., 2006; Naik et al., 2013; Voulgarakis et al., 2013). Several model studies have examined changes in OH abundance and the $CH_4$ lifetime since pre-industrial times. Chemistry-transport models (which use off-line, precalculated meteorology) generally simulate decreases in OH and increases in the $CH_4$ lifetime, ranging from 6% to 25% during the 21$^{st}$ century (Thompson, 1992; Lelieveld et al., 1998; Wild and Palmer, 2008). These results differ from those produced by chemistry-climate models which account for changes in both emissions and climate (Stevenson et al., 2000; Johnson et al., 2001; Shindell et al., 2006; Zeng et al., 2010; John et al., 2012). All of them project a reduction in the $CH_4$ lifetime and an increase in OH. In particular, Shindell et al. (2006) and Zeng et al. (2010) obtain a ∼10% decrease in the $CH_4$ lifetime using different emission scenarios in their simulations. More recent and comprehensive studies compare present-day and future results for OH and the $CH_4$ lifetime among models participating in the Atmospheric Chemistry and Climate Model Intercomparison Project (ACCMIP, Naik et al., 2013; Voulgarakis et al., 2013). Naik et al. (2013) analyse the evolution of the $CH_4$ lifetime and OH in ACCMIP models since preindustrial times (1850-2000). They point out large variations in the sign and magnitude of OH changes (from $-12.7\%$ to $14.6\%$) amongst ACCMIP models, reflecting uncertainties in natural CO, $NO_x$, and NMVOC emissions as well as roles of the diverse chemical mechanisms included in the models. For present-day (year 2000) simulations of OH and the $CH_4$ lifetime, Voulgarakis et al. (2013) suggest that diversity in photolysis schemes and NMVOC emissions might cause large variations in simulated OH and the $CH_4$ lifetime. Trends in OH between 2000 and 2100 are mainly attributed to stratospheric $O_3$ changes and trends in modelled temperature fields.

A useful indirect method for constraining global OH is based on tracking the abundance of long-lived, well-mixed chemicals for which oxidation by OH is the dominant sink and which have a well-quantified, industrial source. The most widely used such species is methyl chloroform ($CH_3CCl_3$) (Prinn et al., 2005; Bousquet et al., 2005; Manning et al., 2005; Krol et al., 2008). Montzka et al. (2011) use $CH_3CCl_3$ measurements to infer only a small interannual variability in OH for 1998-2007. The global multi-model mean OH inferred from the ACCMIP ensemble (Naik et al., 2013) increases slightly ($3.5 \pm 2.2\%$) over the period 1980-2000. This result largely agrees with Montzka et al. (2011) and with other models (Dentener et al., 2003; Hess and Mahowald, 2009; John et al., 2012; Holmes et al., 2013), but disagree with other studies of $CH_3CCl_3$ observations that find a decrease in OH over that period (Prinn, 2001; Bousquet et al., 2005). For the year 2000, Naik et al. (2013) underestimate the $CH_3CCl_3$ lifetime (and thus overestimate OH) by 5% to 10% relative to observations. $CH_3CCl_3$ is controlled under the Montreal Protocol, meaning its abundance in the atmosphere is approaching the detection limit and it will no longer be a useful constraint on OH in decades to come.

A further indirect method to address OH is to measure $^{14}CO$. Manning et al. (2005) find some considerable variability but no long-term trend using this method. According to Krol et al. (2008), this method is considerably more sensitive to high-latitude than low-latitude OH, in contrast to the $CH_3CCl_3$ method which is mostly sensitive to tropical OH.

Therefore, a step forward in addressing the uncertainty in modelling OH in global models is to quantitatively assess the contributions of biases in long-lived species that are central to OH. This sometimes involves juxtaposing global models to local-scale (box or single-column) models constrained as much as possible by observations and incorporating only fast photochemical processes. For example, Emmerson et al. (2005, 2007) develop a box model to assess the sensitivity of OH and
$HO_2$ to biases in long-lived species, and compare the model results to observations. However, their analyses only pertain to polluted environments not representative of much of the global atmosphere and only take in episodic and surface measurements. Single-column models have been applied to modelling the atmospheric boundary layer (Mihailovic et al., 2005; Cuxart et al., 2006), diabatic processes (Randall et al., 2003; Bergman and Sardeshmukh, 2004), clouds and aerosols (Kylling
et al., 2005; Lebassi-Habtezion and Caldwell, 2015; Dal Gesso et al., 2015), the impacts of GHGs on climate change (Vupputuri et al., 1995), and the chemistry of halogen compounds (Piot and von Glasow, 2008; Joyce et al., 2014). Tropospheric OH chemistry of the remote atmosphere has not been assessed in a single-column model framework before.

In the present paper, we introduce and evaluate a single-column model (SCM) constrained with
available long-term observations at Lauder, New Zealand (45°S, 170°E, 370 m above sea level), to investigate how chemistry-climate model biases in long-lived chemical species and temperature affect OH. Lauder is known for its clean air (Stedman and McEwan, 1983; McKenzie et al., 2008) and large diversity of available measurements (it is part of the Network for the Detection of Atmospheric Composition Change (NDACC), Badosa et al., 2007; McKenzie et al., 2008; WMO, 2011).
Observations made at Lauder include UV radiation and surface, profile, and/or total columns of $O_3$ and several other species. The $O_3$, $H_2O$, and temperature records produced by ozone sondes cover 1986 to the present. Lauder therefore is ideal for this kind of study. The SCM is built around a medium-complexity stratosphere-troposphere chemistry scheme. The model is forced with Lauder observations and/or output from a chemistry-climate model that uses the same scheme (see below).
In Section 2, we describe the set-up of the SCM, the construction of time series of key species and meteorological parameters that drive the SCM, and the simulations. In Section 3, we present results of simulated OH concentrations and trends from the SCM and analyse the sensitivity of OH to various forcings. Conclusions are gathered in Section 4.

## 2 Models and simulations

### 2.1 The single-column photochemical model (SCM)

The single-column photochemical model (SCM) is a stand-alone version of the stratosphere-troposphere chemistry mechanism used by the National Institute of Water and Atmospheric Research - United Kingdom Chemistry and Aerosol (NIWA-UKCA) model, which comprises gas-phase photochemical reactions relevant to the troposphere and stratosphere (Morgenstern et al., 2009, 2013; Telford et al., 2013; O'Connor et al., 2014; Morgenstern et al., 2016). For consistency with NIWA-UKCA, the SCM uses the same chemical mechanism. Had we used a more complex mechanism (which the SCM approach lends itself to), then a direct comparison with the NIWA-UKCA output would no longer be possible, and also the results would be less relevant to other global CCMs characterized by relatively simple chemical mechanisms. The 60 vertical levels of the SCM are the same as in NIWA-UKCA, extending to 84 km. We do not use horizontal interpolation and take profiles of atmospheric properties from the gridpoint closest to Lauder ($45°$S,$168.75°$E). Unlike NIWA-UKCA, the SCM excludes all non-chemistry processes, such as transport, dynamics, the boundary-layer scheme, radiation, emissions, etc. This means the model is only suitable for assessing fast photochemistry. Forcing data for the SCM are mostly interpolated from 10-daily instantaneous outputs from a NIWA-UKCA simulation (see below), except for those fields for which observational data are used.

Morgenstern et al. (2013) and O'Connor et al. (2014) describe the chemistry scheme included in the SCM. The SCM includes an isoprene oxidation scheme (Pöschl et al., 2000; Zeng et al., 2008; Morgenstern et al., 2016) not included in the NIWA-UKCA model version used by Morgenstern et al. (2013). In addition to $CH_4$ and CO, the model includes a number of primary non-methane volatile organic compound (NMVOC) source gases, i.e. ethane ($C_2H_6$), propane ($C_3H_6$), acetone ($CH_3COCH_3$), formaldehyde (HCHO), acetaldehyde ($CH_3CHO$), and isoprene ($C_5H_8$). As noted above, emission and deposition of species are not considered in the SCM. The SCM includes a comprehensive formulation of stratospheric chemistry (Morgenstern et al., 2009) comprising bromine and chlorine chemistry and heterogeneous processes on liquid sulfate aerosols. Overall, the model represents 86 chemical species and 291 reactions including 59 photolysis and 5 heterogeneous reactions. The FAST-JX interactive photolysis scheme (Neu et al., 2007; Telford et al., 2013) has been implemented in the SCM; this scheme solves a radiative transfer equation accounting for absorption by ozone. The chemical integration is organised through a self-contained atmospheric chemistry package (Carver et al., 1997), and the differential equations describing chemical kinetics are solved using a Newton-Raphson solver (Morgenstern et al., 2009).

## 2.2 Construction of vertical profiles of forcing species and meteorological parameters

Time series of $O_3, H_2O, CO$, and temperature profiles are produced using mainly long-term measurements from Lauder, supplemented with NIWA-UKCA results as detailed below. Lauder is a member of several international networks, including the NDACC (http://www.ndsc.ncep.noaa.gov), the Global Reference Upper Air Network (GRUAN; http://www.gruan.org), and Global Atmosphere Watch (GAW; http://www.wmo.int/pages/prog/arep/gaw/gaw_home_en.html), where these data are archived and made available. The networks coordinate long-term observations of $O_3$, various other constituents, and meteorological parameters. Here we briefly describe the procedure of constructing forcing data, using Lauder observations, to be used to constrain the SCM. The resulting profiles are shown in Fig. 1.

$O_3$ profiles used here are a combination of ozonesonde time series (from the surface to 25 km, Bodeker et al., 1998) combined with the Microwave Ozone Profiler Instrument 1 (MOPI1) time series for altitudes above 25 km (Boyd et al., 2007; Nedoluha et al., 2015), covering 1994 to 2010 (Fig. 1a). The ozone sondes have been launched approximately weekly; this defines the temporal coverage of the forcing data used in the SCM calculations. Microwave measurements used here come as several profiles a day at a variable spacing; we interpolate them to the ozone sonde launch times. Any missing data (during the two periods when the microwave instrument was out of service) are filled using a Fourier series gap-filling method. We compare the two datasets in the height region usefully covered by both (20 to 30 km). The differences between the two measurements range between $-2\%$ and $+6\%$, and a mean bias that is less than $5\%$. $O_3$ profiles are linearly interpolated onto the SCM's grid. Total column ozone calculated by integrating the observed $O_3$ profiles is also compared to total-column $O_3$ measured by the Lauder Dobson instrument; the difference is about $5\%$ on average (López Comí, 2016). Lauder ozone measurements have been used in various World Meteorological Organization (WMO) ozone assessments (e.g., WMO, 2011).

$H_2O$ profiles have been constructed using the weekly radiosonde measurements of $H_2O$ vapour below 8 km (the same soundings that also measure ozone) and NIWA-UKCA model output data above. For validation, we use the monthly National Atmospheric and Oceanic Administration (NOAA) Frost Point Hygrometer (FPH) $H_2O$ vapour measurements (Vömel et al., 2007; Hall et al., 2016) which start in 2003. FPHs are more accurate compared to radiosonde hygrometers, particularly for stratospheric conditions. However, due to the later start of the FPH time series and the lower measurement frequency, radiosonde measurements are used in this study. The comparison of FPH and radiosonde $H_2O$ reveals differences that are generally less than $\pm5\%$ in the lower and middle troposphere but generally increase in and above the tropopause region ($\sim 11$ km, López Comí, 2016). The radiosonde hygrometers have some known problems with measuring low humidity (Miloshevich et al., 2001). This is reflected in the large differences observed particularly at these altitudes (up to 30%), and to a lesser degree, below them (Fig. 2a). In a comparison of NIWA-UKCA output with FPH $H_2O$, larger discrepancies are found throughout the whole troposphere and tropopause region

(Fig. 2b), as can be expected from a low-resolution model unconstrained by observations and subject
to problems with modelling $H_2O$. Given the consistency of FPH and radiosonde $H_2O$ below 8 km
found before, here we use radiosonde data up to 8 km of altitude merged, in the absence of a more
suitable dataset, with NIWA-UKCA output above that.

We use surface in situ measurements from Cape Grim, Tasmania (Cunnold et al., 2002) to rescale
NIWA-UKCA model profiles, producing $CH_4$ profiles that coincide with the ozone sonde launches.
The NIWA-UKCA model simulation had been constrained with historical global-mean surface $CH_4$
values, resulting in an overestimation relative to the Cape Grim data by $\sim 2\%$ (not shown), and
both data show a $\sim 5\%$ increase in $CH_4$ at the surface over the period between 1994 and 2010. Cape
Grim $CH_4$ is a good surrogate for the Lauder measurements because $CH_4$ is a long-lived, well-mixed
atmospheric constituent.

The time series of CO profile over the period of 1994-2010 has been constructed using the NIWA-
UKCA CO profiles, rescaled such that the total columns match those obtained from the mid-infrared
Fourier Transform Spectrometer (FTS) at Lauder (Rinsland et al., 1998; Zeng et al., 2012; Morgen-
stern et al., 2012). Gaps in the total-column FTS series, such as the period between 1994 and 1996
when the FTS measurements had not started yet, are filled using a regression fit accounting for the
mean annual cycle (modelled as a 6-term harmonic series) and the linear trend.

The time series of temperature profiles are constructed following the same procedure as used in the
construction of $O_3$ profiles, comprising the radiosonde temperature profiles (from the surface to 25
km) merged with NCEP/NCAR reanalyses (Kalnay et al., 1996) temperatures used in the retrieval
of MOPI1 ozone (above 25 km) for the period of 1994-2010. From near the stratopause upwards
the NCEP/NCAR temperatures are merged with a mesospheric climatology based on local LIDAR
measurements. There are some warm anomalies occurring in the data at 40-60 km during winter
months (e.g. in 1996); these may reflect planetary wave breaking in the upper stratosphere.

### 2.3 Simulations

We perform SCM simulations covering the period of 1994-2010, summarized in Table 1. The forcing
data needed by the SCM consist of profile series of temperature, pressure, optionally cloud liquid
and ice mass mixing ratios, and the mixing ratios of 86 chemical compounds. With the exceptions
detailed below, these fields and species are taken from a NIWA-UKCA simulation for the period of
1994-2010 interpolated to the times of the ozone sonde launches. The CCM simulation used here
consists of the last 17 years of a NIWA-UKCA "REF-C1" simulation conducted for the Chemistry-
Climate Model Initiative (CCMI; Eyring et al., 2013). REF-C1 is a hindcast simulation for the period
of 1960 to 2010, using prescribed Hadley Centre sea Ice and Sea Surface Temperature (HadISST)
fields (Rayner et al., 2003). The surface emissions of primary species are as described in Eyring et al.
(2013), ozone-depleting substances (ODSs) follow the A1 scenario of the World Meteorological
Organisation (WMO) Report (WMO, 2011), and surface (or bulk, in the case of $CO_2$) abundances

of greenhouse gases (GHGs) follow the "historical" Intergovernmental Panel on Climate Change (IPCC) scenario of global-mean GHG mixing ratios (Meinshausen et al., 2011).

In a "reference" simulation of the SCM all forcings are taken from this REF-C1 simulation of NIWA-UKCA. Alternatively, in sensitivity simulations $O_3, H_2O, CH_4, CO$, and temperature, or all of these simultaneously, are replaced with the time series of the profiles that are constructed using 240 long-term observational data as described above. For species other than those 5 fields, in all cases we use NIWA-UKCA REF-C1 forcings. We evaluate the SCM only for those times, spaced roughly weekly, for which ozone sonde data are available. With the exceptions of those simulations assessing cloud influences, simulations are conducted assuming clear-sky conditions.

## 3   OH sensitivity to correcting chemistry-climate model biases

In this section, we present sensitivity studies to assess the contribution of biases in known factors ($O_3, H_2O, CH_4, CO$, and temperature) affecting OH photochemistry at Lauder. The response of OH to changes in each forcing is assessed individually and also in combination.

### 3.1   OH sensitivity to $O_3$ biases

Three sensitivity simulations are conducted to quantify the impact of $O_3$ biases (defined as differ-
250 ences between observed $O_3$ and NIWA-UKCA simulated $O_3$) on OH at Lauder.

As discussed above, the rate of production $P$ of $HO_x$ via $O(^1D) + H_2O$ can be expressed as

$$P(\mathrm{HO_x}) \approx \frac{2k_1 j_{O1D}[\mathrm{O_3}][\mathrm{H_2O}]}{k_2[\mathrm{O_2}] + k_3[\mathrm{N_2}] + k_1[\mathrm{H_2O}]} \tag{1}$$

where $k_1$ is the rate coefficient for $O(^1D) + H_2O$, $j_{O1D}$ is the rate of $O_3$ photolysis producing $O(^1D)$, and $k_2$ and $k_3$ are the rate coefficients of quenching of $O(^1D)$ with $O_2$ and $N_2$, respectively 255 (Liu and Trainer, 1988; Thompson et al., 1989; Madronich and Granier, 1992; Fuglestvedt et al., 1994). Accordingly, $P(\mathrm{HO_x})$ is affected by ozone changes principally in two different ways: Either locally through a change in $[O_3]$ or non-locally through a change in $j_{O1D}$ caused by changes in the overhead total-column ozone (TCO). To separate the effects, we conduct three simulations with the SCM: The first simulation targets the local kinetics effect by applying changes in $O_3$ concentrations 260 but keeping all photolysis rates unchanged versus the reference simulation. A second simulation involves applying changes in $j_{O1D}$ according to changes in $O_3$ (keeping the rest of photolysis rates unchanged), but considering a fixed $O_3$ concentration, i.e. using the $O_3$ concentrations of the reference simulation. The $j_{O1D}$ calculation consistently takes into account absorption and scattering by stratospheric and tropospheric $O_3$. A third simulation includes both effects simultaneously.

The results of these three sensitivity runs are displayed in Fig. 3. As expected, the pattern of $O_3$ differences between observed $O_3$ and modelled $O_3$ (Fig. 3a) is reflected in the pattern of OH differences produced by the SCM, considering only the "kinetics" effect and assuming no changes

in the photolysis rates (Fig. 3b), with increases of ozone in spring and decreases in autumn, relative to the reference simulation, resulting in changes of the same sign in OH. However, there is a height dependence to this relationship. In summer and autumn, $O_3$ biases range between $-5\%$ and $-45\%$, meaning that the reference simulation overestimates the observations. Such a bias in $O_3$ results in up to 12% reductions in OH for these seasons when the bias is corrected. In spring between 2 and 6 km, observed $O_3$ is larger than in the reference simulation by up to 10% at 4 km in October. Consequently, this results in an increase of OH at around the same altitudes and times of up to 5%.

Regarding the sensitivity simulation considering the photolysis effects, $j_{O1D}$ exhibits differences relative to the reference simulation ranging from $\sim 14\%$ to $\sim 30\%$. The corrections are positive everywhere, in accordance with the overestimation of TCO in the NIWA-UKCA model with respect to observations (Morgenstern et al., 2013; Stone et al., 2016). In accordance with eq. 1, such an intensification of $j_{O1D}$ causes OH to increase (Fig. 3c). The relative OH response is approximately 50% of the $j_{O1D}$ relative difference. However, Figs. 3(c) and (d) suggest that the magnitudes of the kinetics and the photolysis effects, for the $O_3$ bias found at Lauder, are comparable, but the seasonalities differ. For example, the kinetics effect maximizes in spring at 5% and minimizes in summer/early autumn at $-15\%$ (in the upper troposphere) whereas the photolysis effect on OH maximizes in summer at 16 to 20% and minimizes in spring (figs. 3b and 3d).

OH resulting from the combined kinetics and photolysis effects is displayed in Fig. 3(e). OH responds approximately linearly to the two effects combined, compared to the sum of their individual impacts (Fig. 3f), despite some small differences between Fig. 3(e) and (f).

Next, we examine the relationship between slant column of $O_3$ (SCO), $j_{O1D}$, and OH. Fig. 4(a) shows that there is an approximately exponential relationship between $j_{O1D}$ and the SCO at 6 km of altitude (this effect also exists at other altitudes). The small curvature may be the result of inaccurately diagnosing the SCO (ignoring the curvature of the Earth). Another reason could be that the cross section of $O_3$ is wavelength dependent, and consequently the actinic flux spectrum moves towards longer wavelengths with increasing SCO. Under Lambert-Beer's Law, a perfectly exponential relationship would be expected for a monochromatic UV light source and an isothermal atmosphere. $j_{O1D}$ and the OH concentration exhibit an approximately linear relationship ( eq. 1, fig. 4b). Combining these results, we derive an approximately exponential relationship between the SCO and the OH concentration (fig. 4c). The fit parameters are stated in fig. 4. Due to the compact relationship between $j_{O1D}$ and the SCO, and to account for the curvature, we fit a quadratic relationship between the SCO and $\log(j_{O1D})$.

To determine a simple coefficient that describes the quantitative contribution of $O_3$ to OH, a linear regression between differences in OH and $O_3$ relative to the reference was conducted through the following expression (note that this equation is also used to derive the linear contributions of the other key species to OH chemistry at Lauder):

$$\frac{\Delta[\text{OH}]}{[\text{OH}]_{ref}} = A_i \frac{\Delta X_i}{X_{i,ref}} \tag{2}$$

where $X_i$ is the perturbation variable (in this case $[O_3]$), $A_i$ is the slope of the linear regression, $\Delta[OH]$ is the absolute difference between the OH concentrations in the reference and perturbation simulations, $\Delta X$ is the absolute difference in concentrations of the perturbation variable $X$ between the observations and the reference, $[OH]_{ref}$ is the OH concentration obtained from the reference simulation, and $X_{i,ref}$ is the value of $X_i$ in the reference simulation. The regression coefficients $A_i$ represent the sensitivity of OH to changes in each individual variable for the troposphere at Lauder. The regression coefficients are depicted in Fig. 5. Reverting to infinitesimal notation, we note that

$$A_i = \frac{\partial \ln[OH]}{\partial \ln X_i}. \tag{3}$$

The sensitivity coefficients of OH to the kinetics and photolysis effects of $O_3$ are shown in Fig. 5(a). Coefficient $A_1$, which represent the kinetics effect, ranges from $0$ to $0.25$ (meaning the relative response of OH is up to a quarter of the relative difference in $O_3$). The sensitivity to photolysis ($A_1''$) is $> 0.5$ throughout much of the troposphere (meaning the relative response in OH is over half the relative change in $j_{O1D}$).

### 3.2 OH sensitivity to $H_2O$ biases

A perturbation simulation was performed using combined radiosonde and NIWA-UKCA $H_2O$ (section 2.2). The OH response to correcting $H_2O$ biases (Fig. 6) shows an approximately linear response with respect to the relative changes in $H_2O$, i.e. decreases in $H_2O$ generally lead to a reduction of OH concentrations (eq. 1). Note that NIWA-UKCA substantially overestimates the radiosonde-observed $H_2O$ vapour by up to 60% between 2 and 6 km (Fig. 6a); this translates into an overestimation of OH by up to $\sim 40\%$ in the reference simulation (Fig. 6c) in the same region. The sensitivity of OH to changes in $H_2O$ (eq. 2) range from $5$ to $0.5$ in the troposphere (Figs. 6e and 5 (b) coefficient $A_2$), with high sensitivity in the lower and free troposphere and reduced sensitivity in the tropopause region.

It is known that large uncertainties are associated with $H_2O$ vapour measurements. To illustrate this, we repeat the above simulation but now using European Centre for Medium–Range Weather Forecasts (ECMWF) ERA–Interim reanalysis (hereafter ERAI) $H_2O$ (Dee et al., 2011). Irrespectively of the large differences and the opposite signs in $H_2O$ biases between Lauder radiosonde and ERAI data, the OH response to biases in $H_2O$ show approximately the same linear relationship in both cases (Fig. 6). Likewise, the sensitivity of OH to changes in $H_2O$ using ERAI data (Figs. 6f and 5b, coefficient $A_3$) resembles the sensitivity simulation using radiosonde $H_2O$.

### 3.3 OH sensitivity to $CH_4$ and CO biases

The effect of $CH_4$ changes on OH is displayed in Fig. 7 (a,c,e). The $CH_4$ biases are generally small, up to only $\sim 2\%$, and are assumed to be vertically uniform, with some seasonal variations. Decreases in $CH_4$ lead to increases in OH due to reduced loss of $HO_x$ by $CH_4 + OH$. The response of OH

to $CH_4$ changes maximizes at 0.6% around 2 km, and decreases at higher altitudes. The seasonal
variation of the OH response to $CH_4$ biases maximizes in March/April (Fig. 7c), which coincides
with the maximum absolute bias in $CH_4$ (Fig. 7a) in the same months. The sensitivity coefficient
describing the dependence of OH to $CH_4$ changes (denoted as $A_4$ in Fig. 5c) ranges from $-0.17$ at
the surface to $-0.32$ at $\sim 2$ km of altitude, and then decreases to $-0.15$ at 10 km.

The CO bias and the resulting differences in OH are displayed in Fig. 7(b,d,f). The relative dif-
ference of OH with respect to the reference simulation is less than $\pm 5\%$ for all seasons (Fig. 7d),
showing that decreases in CO generally lead to increases in OH through the reduced loss of OH
through $OH + CO$. Note that during austral spring NIWA-UKCA overestimates CO, presumably
due to exaggerated tropical biomass burning in the model which causes CO biases of up to 10%
(Fig. 7b). The sensitivity of OH to changes in CO ($\partial \ln[OH]/\partial \ln[CO]$) shown in Fig. 7(f) varies
from $-0.3$ to $-0.5$ and in absolute terms increases with altitude (the white band shown in October
is the result of CO differences being close to zero).

The sensitivities of OH to $CH_4$ and CO show comparable values at the surface, but the OH sensi-
tivity to CO increases with height whereas its sensitivity to $CH_4$ decreases. Note that the $CH_4 + OH$
reaction rate is strongly temperature dependent, which may contribute to the lower sensitivity of OH
to $CH_4$ changes at altitude than to CO. However, further investigation will need to investigate how
these ratios change in different chemical regimes, and to assess whether the relative sensitivity of
OH to CO and to $CH_4$ are specific to the clean SH environment.

### 3.4 OH sensitivity to temperature biases

To assess the effects of changes in temperature on OH, we apply the same procedure as for $O_3$,
for which the effects of temperature have been decomposed into kinetics and photolysis effects. We
perform three simulations: In the first simulation, we only apply temperature changes to chemical
kinetics, keeping all photolysis rates fixed (noting that most uni-, bi-, and termolecular reaction rates
are temperature dependent). In the second simulation, we only consider the photolysis effect, which
arises mainly because the cross section of $O_3$, the primary UV absorber, is temperature dependent.
The impact of temperature on OH via ozone photolysis again occurs via two different mechanisms:
firstly, the changes in $j_{O1D}$ caused by changes in the actinic flux which relates to changes in the
atmospheric transmissivity in the UV (caused by a temperature dependence of the cross section of
overhead ozone), and the local changes of $j_{O1D}$, due to the local temperature dependence of the
ozone cross section. Here, we only evaluate the combined photolysis effect in the second simula-
tion. Finally, we perform a third simulation by applying both the kinetics and the photolysis effects
simultaneously.

At Lauder, the reference simulation is generally cold-biased (i.e., the temperature correction is
positive; fig. 8a). This is particularly the case in the lowest 2 km and throughout the troposphere
in the autumn-winter season. The kinetics effect leads to a reduction of OH by up to 2% (fig. 8b).

$O(^1D) + H_2O$ and the quenching reactions (eq. 1) are not or weakly temperature dependent, making $CH_4 + OH$ (which is much more sensitive to temperature) the leading factor in causing this small OH reduction. The rate coefficient for this reaction in NIWA-UKCA and the SCM is $k_{OH+CH4} = 1.85 \cdot 10^{-12} \exp(-1690\mathrm{K}/T)$; at 290 K the sensitivity of $k_{OH+CH4}$ to temperature changes evaluates to about 2%/K. However, OH is well buffered by other reactions, so its sensitivity is considerably

smaller than that. The photolysis effect is often somewhat larger than the kinetics effect but peaks in spring (fig. 8c). This translates into a slight OH reduction comparable in magnitude to the kinetics effect (fig. 8d). Both effects add nearly linearly in the combined simulation (fig. 8e,f).

     We calculate sensitivity coefficients $A_6$ and $A_6''$ that define the OH responses to both effects (fig. 5 e,f). Coefficient $A_6$ represents the kinetics effect and varies from 0 to $-1.75$ (i.e., in absolute terms,

the relative OH response can be larger than the relative difference in $T$). The sensitivity coefficient that describes the sensitivity of OH to changes in photolysis ($A_6''$) ranges from 0.6 at the surface to 0 at 10 km of altitude. Figure 5 (e,f) shows sensitivity coefficients for both effects ($A_6$ and $A_6''$). OH changes due to both effects are small (up to 2.5%) and comparable in magnitude.

     Several sensitivity studies have been conducted previously to elucidate the impact of temperature

on OH (Stevenson et al., 2000; Wild, 2007; O'Connor et al., 2009). None of these studies separately assessed the impacts of the kinetics and photolysis effects of temperature on OH. Wild (2007) applied a globally uniform temperature rise of 5 K that led to a larger OH abundance and an around 10% decrease in the $CH_4$ lifetime. O'Connor et al. (2009) showed a small impact on global OH abundances due to temperature biases; this may be because either the temperature biases in their

model were both positive and negative, in different regions, leading to some cancellation of the impact on global OH, or to low OH sensitivity to temperature biases. Here, bias-correcting temperature is shown to also have only a small impact on OH abundance (Fig. 8e); this result broadly corroborates that of O'Connor et al. (2009).

### 3.5    Linearity of OH sensitivity to biases in all forcings

Here, we assess the effect of changing all forcings ($O_3$, $H_2O$, $CH_4$, $CO$, and temperature) simultaneously on OH at Lauder. Fig. 9(a) shows the responses of OH to changing all forcings. A comparison with fig. 6 suggests that $H_2O$ changes dominate the total response of OH to changes in these forcings. At Lauder, NIWA-UKCA is too moist (relative to radiosonde water vapour); this translates into a large OH overestimation of up to $\sim 40\%$ in the reference simulation (Fig. 9a). This is consis-

tent with the underestimated $CH_4$ lifetime by the NIWA-UKCA model (Morgenstern et al., 2013; Telford et al., 2013), assuming that the NIWA-UKCA model is too moist also in other regions. (In the NIWA-UKCA reference simulation used here, the global $CH_4$ lifetime, disregarding dry deposition, is 7.2 years, whereas a recent best estimate is 9.8 years, with an uncertainty range of 7.6 - 14 years (SPARC, 2013)). In general, in the SCM OH responds approximately linearly to the combined

changes in major forcings that play an important role in OH chemistry (Fig. 9).

To examine the linearity of OH responses to simultaneous changes in key forcings defined in this study, the combination of all individual contributions, i.e. $O_3$ (kinetics and photolysis effects), $H_2O$, $CH_4$, CO, and temperature (kinetics and photolysis effects) to OH, was compared to the OH response to all forcings combined simulation in the SCM through Eq. (4):

$$
\frac{\Delta[\text{OH}]}{[\text{OH}]_{\text{ref}}} \approx A_1 \frac{\Delta[\text{O}_3]}{[\text{O}_3]_{\text{ref}}} + A_1'' \frac{\Delta j\text{O}(^1\text{D})_{\text{O}_3}}{j\text{O}(^1\text{D})_{\text{ref}}} + A_2 \frac{\Delta[\text{H}_2\text{O}]}{[\text{H}_2\text{O}]_{\text{ref}}} + A_4 \frac{\Delta[\text{CH}_4]}{[\text{CH}_4]_{\text{ref}}} + A_5 \frac{\Delta[\text{CO}]}{[\text{CO}]_{\text{ref}}}
$$
$$
+ A_6 \frac{\Delta T}{T_{\text{ref}}} + A_6'' \frac{\Delta j\text{O}(^1\text{D})_{\text{T}}}{j\text{O}(^1\text{D})_{\text{ref}}}, \tag{4}
$$

where $\Delta[\text{OH}]/[\text{OH}]_{\text{ref}}$ is the relative difference in the OH concentration obtained with the SCM with respect to the reference simulation, using all forcings combined. The forcings comprise the kinetics and photolysis effects of $O_3$ ($A_1$ and $A_1''$), radiosonde $H_2O$ ($A_2$), $CH_4$ ($A_4$), CO ($A_5$), and the kinetics and photolysis effects of temperature ($A_6$ and $A_6''$). Eq. 4 expresses a working hypothesis that the model responds linearly to the applied forcings; we will test this assumption in the following paragraph.

Figure 9 a,b indicates that the model responds approximately linearly to the combinations of all forcings, with OH responses in the all-forcings simulation correlating at 0.9 with the sum of the OH responses in the individual-forcing simulations. Fig. 9(c) however also suggests that there are some notable non-linearity in the chemistry of the troposphere at Lauder. Chemical feedbacks between the impacts of correcting water vapour and ozone may contribute to this non-linearity; for example, a change in the water vapour abundance may impact the sensitivity of OH to changing $O_3$.

### 3.6 Trends in OH

We examine variability and trends in OH using the SCM simulation including all key forcings separately for different altitude bins. The results (Fig. 10) indicate that there are no significant long-term trends in OH throughout the troposphere for the period of the simulation (1994-2010) We find trends of $-2.1 \pm 4.8\%$ at 0-2.5 km, $0.9 \pm 2.3\%$ at 2.5-5 km, $2.6 \pm 3.5\%$ at 5-7.5 km, and $3.6 \pm 4.1\%$ at 7.5-10 km over the period of 1994-2010), but there is evidence of interannual variations at all altitudes (e.g., Manning et al., 2005; Montzka et al., 2011).

In addition, we explore variability and trends in the OH column at Lauder to be compared with other estimates of global OH. As expected from the results of OH trends at different altitude bins, we find no significant long-term trend in the OH column ($0.5 \pm 1.3\%$) (Fig. 11). However, there is evidence of short-term variations ($5 - 10\%$), in agreement with other studies that used observations to infer global OH concentrations (e.g., Manning et al., 2005; Montzka et al., 2011).

### 3.7 OH sensitivity to the presence of clouds

We have assessed the OH sensitivity to correcting biases in key forcings assuming clear skies. Here we explore the impact of simulated clouds on OH, recognizing that this process is associated with large uncertainties due to difficulties with representing clouds in models. Measurements of cloud

profiles do not exist at Lauder, hence a bias correction like that performed with the composition and temperature fields is not possible. Therefore, here we only examine the impact of clouds simulated by NIWA-UKCA on $j_{O1D}$ and OH at Lauder, relative to the clear-sky reference simulation used before. The impacts of liquid water clouds (LWCs) and ice clouds (ICs) were assessed separately and in combination. Three simulations are defined here, i.e. (1) including only ICs, (2) including 450 only LWCs, and (3) considering both combined (LICs).

Fig. 12(a,c,e) shows the response of $j_{O1D}$ and OH to the presence of the ICs. $j_{O1D}$ and OH are generally reduced below the ICs, relative to the cloud-free situation. The maximum reduction in OH is 10 to 15% in winter below 2 km, coinciding with the maximum reduction in $j_{O1D}$. There are increases in both fields (up to $\sim 8\%$) above the ICs in austral spring, associated with the seasonal 455 peak in IC occurrence at the same time. In general, $j_{O1D}$ and OH impacts vary strongly with season, with the maximum reduction occurring in winter close to the surface, and the maximum increase in spring above the ICs.

LWCs are mostly present between 1 and 4 km with the seasonal peak in austral spring (fig. 12b). Similarly to ICs, $j_{O1D}$ and OH are enhanced above and throughout much of the cloud layer, and 460 reduced in the lowest 1 km above the surface (fig. 12e,g). The enhancement in $j_{O1D}$ and OH peaks at 12% between 2 and 4 km of altitude, coinciding with the spring maximum in liquid water content at 1–2 km. Conversely, the reduction in $j_{O1D}$ and OH with respect to the clear–sky condition is $\sim 10\%$ and is produced below the clouds.

The simulation with the combined effect of ICs and LWCs (LICs) produces a reduction in $j_{O1D}$ 465 and OH that ranges between 0% and 20% below the transition of ICs to LWCs at around 2 km, since LWCs are as much as twice as optically dense as ICs (fig. 12g). An enhancement is produced above this altitude of up to 18%. The magnitudes of changes in $j_{O1D}$ and OH are similar when either ICs or LWCs are considered in the SCM. Furthermore, their effects add up slightly less than linearly when both are present in the simulations (fig. 12h).

The results shown here indicate that lower clouds generally produce an enhancement in $j_{O1D}$ (Fig. 12d), but higher clouds generally produce a reduction in $j_{O1D}$ in the free troposphere (Fig. 12b; Tang et al., 2003; Tie et al., 2003; Liu et al., 2009). Furthermore, the vertically and seasonally averaged enhancement and reduction in $j_{O1D}$ are about 2% and 6% respectively for the LWC clouds, similar to the response for the ICs condition; this suggests that the cloud vertical distribution has a 475 bigger effect on photolysis than the change in cloud water content (Tie et al., 2003).

## 4  Conclusions

The sensitivity of the OH abundance at Lauder to NIWA-UKCA model biases in key forcing variables ($O_3, H_2O, CH_4, CO$, and temperature) have been quantified for clear-sky conditions, using a single-column model (SCM). Only fast photochemistry is represented in the SCM; slow chemistry

(i.e. timescales similar to or longer than the 1-hour chemical timestep), transport, and other physical processes are thus not considered. The bias-corrected profiles of the key forcing variables have been constructed largely using long-term Lauder measurements, combined with NIWA-UKCA output. Also a few other sources of data (Cape Grim methane measurements, ERA-Interim water vapour) have been used.

The results show that OH responds approximately linearly to correcting biases in $O_3, H_2O, CH_4$, CO, and temperature. We have decomposed the OH response to $O_3$ changes into the kinetic effect (i.e. local impacts on the chemical steady state of changing $[O_3]$) and the photolysis effect (as mediated by changes in the overhead $O_3$ column affecting photolysis rates). We find that the kinetic effect of correcting positive biases in modelled $O_3$ causes a reduction in OH during austral summer and

autumn (by up to $20\%$ at 7 km), and an increase in the free troposphere in austral spring (of $> 5\%$ in October at 3 km); such changes in OH are nearly linearly related to the corresponding ozone biases. NIWA-UKCA generally overestimates the ozone column. Correcting this bias causes $j_{O1D}$ to increase by $15 - 30\%$ below 10 km, causing general OH increases which maximize at around $16\%$ between 2 and 6 km in summer. The model responds approximately linearly to the combined effects

of photolysis and kinetics.

    NIWA-UKCA considerably overestimates the $H_2O$ vapour concentration by up to $\sim 50\%$ compared to radiosonde measurements. Correcting this moist bias leads to $> 34\%$ reductions in OH in the free troposphere during the austral summer. The sensitivity coefficient of OH to biases in $H_2O$ vapour is relatively large in the lower troposphere but decreases with altitude. Assuming this moist

bias is not restricted to Lauder (which we do not assess here), this is thus a leading explanation for NIWA-UKCA to produce an underestimated $CH_4$ lifetime (Morgenstern et al., 2013; Telford et al., 2013), relative to literature estimates (Naik et al., 2013; Voulgarakis et al., 2013; SPARC, 2013).

    The bias in modelled $CH_4$ is small since surface $CH_4$ in the SCM reference simulation is constrained to follow globally averaged surface observations. The Southern Hemisphere generally has a

slightly smaller $CH_4$ burden than the North. Correcting the resulting positive bias at Lauder causes increases in OH throughout the troposphere, with a seasonal peak in March/April. OH is most sensitive to $CH_4$ changes in winter, though. In the analysis of the OH sensitivity to $CH_4$, the impact of subsequent changes in $CH_4$ oxidation products which also affect OH could not be addressed within the constraints of an SCM. Inclusion of this effect could change the sensitivity coefficient for $CH_4$

(Spivakovsky et al., 2000).

    Except for October-December, NIWA-UKCA has a tendency to underestimate CO. As with $CH_4$, the sensitivity of OH to changes in CO is negative throughout the troposphere, reflecting that $CO + OH$ is an important sink for OH.

    We show that OH responds linearly to temperature biases. These effects cause a reduction in

OH due to the strong dependence of $OH + CH_4$ on temperature (eq. 1). However, the impact of

this reaction on OH is buffered by other less temperature-dependent reactions, causing only a small sensitivity of OH to temperature. This is in agreement with O'Connor et al. (2009).

The results of the simulation considering simultaneous changes in all the key forcings indicate that OH responds approximately linearly to all the major forcings that contribute to the oxidising capacity of the atmosphere. We find that biases in $O_3, H_2O, CH_4, CO$, and temperature all affect the oxidising capacity of the atmosphere at Lauder, with $H_2O$ and $O_3$ biases dominating. We find no significant trend in OH over Lauder over the period 1994-2010.

The SCM approach can be applied to other parts of the globe where reliable long-term observations of $O_3$ and $H_2O$ exist. In-situ observations of $CH_4$ and CO are not that critical; $CH_4$ can be estimated from non-local measurements, and relatively reliable satellite measurements of total-column CO exist (e.g., Pan et al., 1995; Morgenstern et al., 2012). However, in polluted regions, such as in much of the Northern Hemisphere, $NO_x$ and NMVOC levels are elevated relative to Lauder and affect in situ ozone production. This means that these constituents might need to be bias-corrected if the SCM is applied in such regions. This might affect the suitability of our approach under these conditions.

Having determined the contributions of the major forcings to the chemistry of OH at Lauder under clear-sky conditions, a step forward would be to assess the impact of clouds on photolysis and thus OH, which could be substantial. Due to a lack of suitable observations to constrain the SCM model with cloud profiles at Lauder, we only assessed how the presence of modelled cloud affects OH, relative to the clear-sky situation. The results show that OH response to cloud strongly depends on the vertical distribution of the clouds, not just the total amount. Both liquid- and ice clouds lead to increases in OH above and to some extent inside the cloud, particularly in the spring season when this effect maximizes. Considering that clouds are amongst the most difficult aspects of the climate system to model adequately, we stipulate that observational profiles of cloud properties would be highly desirable to use for a future continuation of this line of research.

In summary, we conclude that at Lauder, OH modelled in NIWA-UKCA is most sensitive to issues with representing water vapour and ozone. This points to the need to improve representations of the hydrological cycle and of tropospheric and stratospheric ozone chemistry in NIWA-UKCA and possibly other, similar chemistry-climate models. Water vapour is coupled to clouds in NIWA-UKCA; it is well known that clouds are difficult to represent adequately in global low-resolution climate models. The biases in ozone may well be partly caused by the moist bias in NIWA-UKCA; this is a subject of ongoing research.

Progress with the simulation of the hydrological cycle in present-generation Earth System Models should improve the simulated water vapour product. Simulating an accurate hydrological cycle has been a long-standing issue in climate models, and progess has been slow. If errors in the simulation of moisture cannot be avoided, perhaps their impact on OH can be corrected for using an approach similar to that which we have presented but using global water vapour measurements.

Such a "correction" of modelled OH might result in a reduction in the inter-model spread of the OH abundance and consequently a more accurate quantification of the methane lifetime. For this, tropical radiosonde data would be particularly valuable – most OH is located in the tropics (SPARC, 2013). A similar approach could be used to account for the influence of errors in ozone, although tropospheric in situ ozone measurements may be too sparse to allow for a sufficient characterization of the error in models.

**Author contributions**

O. Morgenstern devised the original idea. L. López Comí wrote the model, conducted the simulations, performed the data analysis, and led the writing of the paper, with support from S. Masters, O. Morgenstern, and G. Zeng. G. Nedoluha contributed the microwave ozone data to the research; R. Querel contributed the ozone sonde data. All authors contributed to the writing of the manuscript.

*Acknowledgements.* All data used in this paper can be obtained from the contact author. This work has been supported by NIWA as part of its Government-funded, core research from New Zealand's Ministry of Business, Innovation, and Employment (MBIE). We would like to thank the Lauder team for providing most of the measurements used here. We particularly thank Dan Smale for his help with various aspects of this work. We acknowledge NOAA for the FPH data. We acknowledge ECMWF for provision of the ERA-Interim data and NOAA/OAR/ESRL PSD, Boulder, Colorado, USA, for provision of the NCEP/NCAR reanalyses. We acknowledge CSIRO Marine and Atmospheric Research and the Australian Bureau of Meteorology for the Cape Grim methane measurements. CSIRO and the Australian Bureau of Meteorology give no warranty regarding the accuracy, completeness, currency or suitability for any particular purpose and accept no liability in respect of data. We acknowledge the U.K. Met Office for use of the MetUM. Furthermore, we acknowledge the contribution of NeSI high-performance computing facilities to the results of this research. NZ's national facilities are provided by the NZ eScience Infrastructure and funded jointly by NeSI's collaborator institutions and through MBIE's Infrastructure programme (https://www.nesi.org.nz).

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

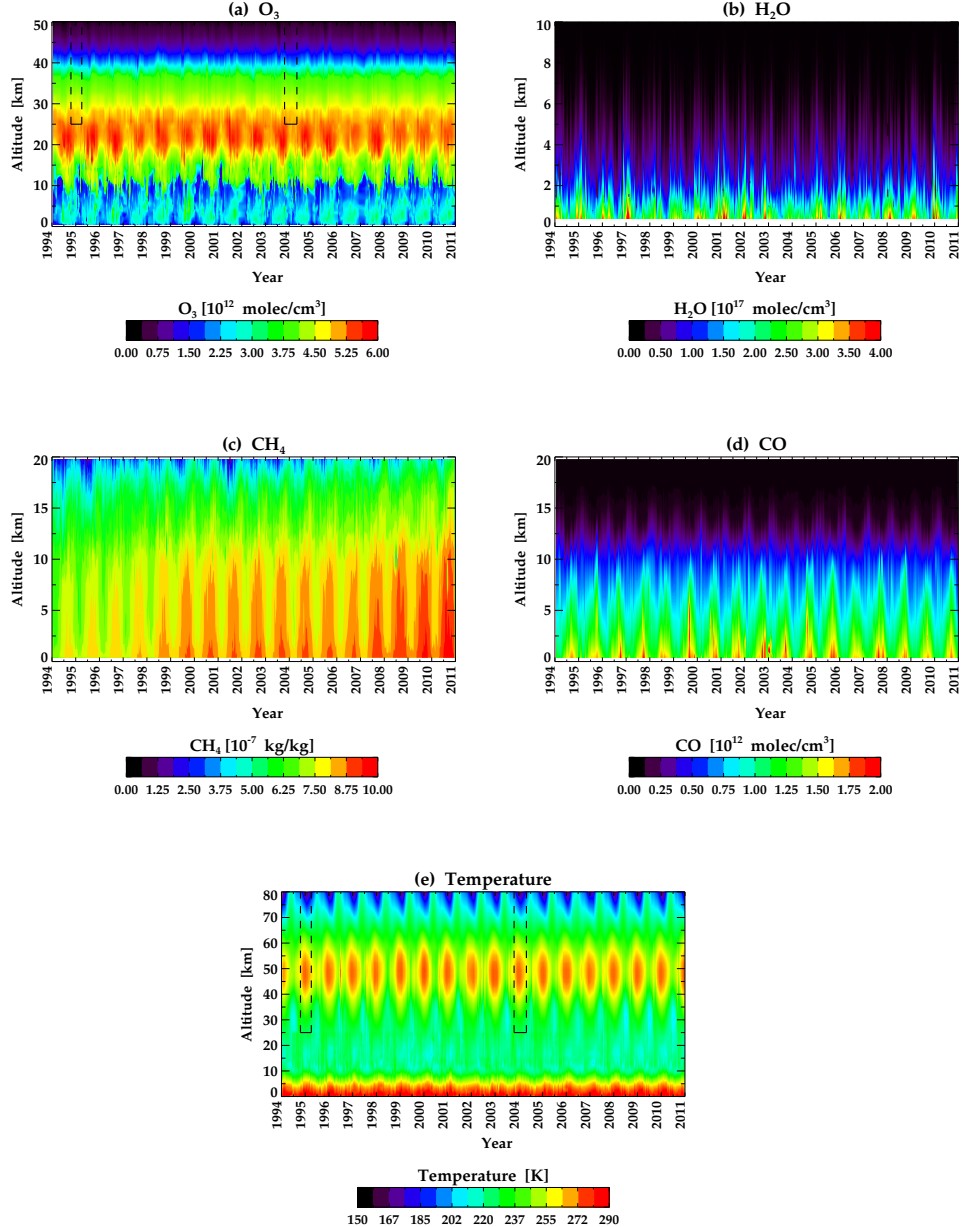

**Figure 1.** (a) Time series of $O_3$ profiles constructed by ozonesonde measurements spliced with MOPI1 measurements. (b) Time series of $H_2O$ profiles constructed by radiosonde measurements spliced with NIWA-UKCA $H_2O$ (time series of ERAI – NIWA-UKCA $H_2O$ is not displayed here). (c) Time series of $CH_4$ profiles constructed by rescaling the NIWA-UKCA $CH_4$ to surface $CH_4$ measurements from Cape Grim (Tasmania). (d) Time series of $CO$ profiles constructed by rescaling the NIWA-UKCA $CO$ to $CO$ measurements from the FTIR spectrometer. (e) Time series of temperature profiles constructed by radiosonde measurements (up to 25 km) merged with NCEP/NCAR reanalyses up to the stratopause (50 km) and a mesospheric climatology based on local LIDAR measurements. Above 25 km these data are as used in the retrieval of $O_3$ from MOPI1 measurements. The areas within black boxes were filled using a Fourier series gap filling method.

| Forcings | Data used |
|---|---|
| $O_3$ | 1. Kinetics effect: $O_3$ changes $\rightarrow$ ozonesondes (0-25 km) + MOPI1 (26-84 km)<br><br>NIWA-UKCA data for other species and temperature<br><br>2. Photolysis effect: $j_{O1D}$ changes according to $O_3$ changes<br><br>NIWA-UKCA data for all species and temperature<br><br>3. Kinetics + photolysis effects: $O_3$ changes –> ozonesondes + MOPI1<br><br>$j_{O1D}$ changes according to $O_3$ changes<br><br>NIWA-UKCA data for other species and temperature |
| $H_2O$ | 1. Changes in $H_2O$ –> radiosondes (0-8 km) + NIWA-UKCA $H_2O$ (9-84 km)<br><br>NIWA-UKCA data for other species and temperature<br><br>2. Changes in $H_2O$ –> ERAI (0-8 km) + NIWA-UKCA $H_2O$ (9-84 km)<br><br>NIWA-UKCA data for other species and temperature. |
| $CH_4$ | Changes in $CH_4$ –> rescaled NIWA-UKCA $CH_4$ to Cape Grim surface $CH_4$<br><br>NIWA-UKCA data for other species and temperature. |
| CO | Changes in CO –> rescaled NIWA-UKCA CO profiles to FTIR CO<br><br>NIWA-UKCA data for other species and temperature |
| $T$ | radiosondes ( surface-25 km) +<br><br>1. Kinetics effect: temperature changes –> NCEP/NCAR reanalyses (26-50 km) +<br><br>LIDAR climatology (50-84 km)<br><br>NIWA-UKCA data for all species<br><br>2. Photolysis effect: $j_{O1D}$ changes according to temperature changes<br><br>NIWA-UKCA data for all species and temperature<br><br>radiosondes ( surface-25 km) +<br><br>3. Kinetics + photolysis effects: temperature changes –> NCEP/NCAR (26-50 km) +<br><br>LIDAR climatology (50-84 km)<br><br>$j_{O1D}$ changes according to temperature changes<br><br>NIWA-UKCA data for all species |
| $O_3$, $H_2O$,<br>$CH_4$, CO, $T$ | Changes in $O_3$, $H_2O$, $CH_4$, CO, and temperature using observations mentioned above.<br>For $H_2O$, radiosonde (0-8 km) + NIWA-UKCA (9-84 km) data are used. |
| Reference | NIWA-UKCA data for all species and temperature |

**Table 1.** Simulations performed with the SCM to assess the contribution of changes in the key forcings to OH chemistry at Lauder under clear-sky conditions. The table includes the type of measurement/data set used to prescribe the key forcings. The time period of simulation is between 1994 and 2010.

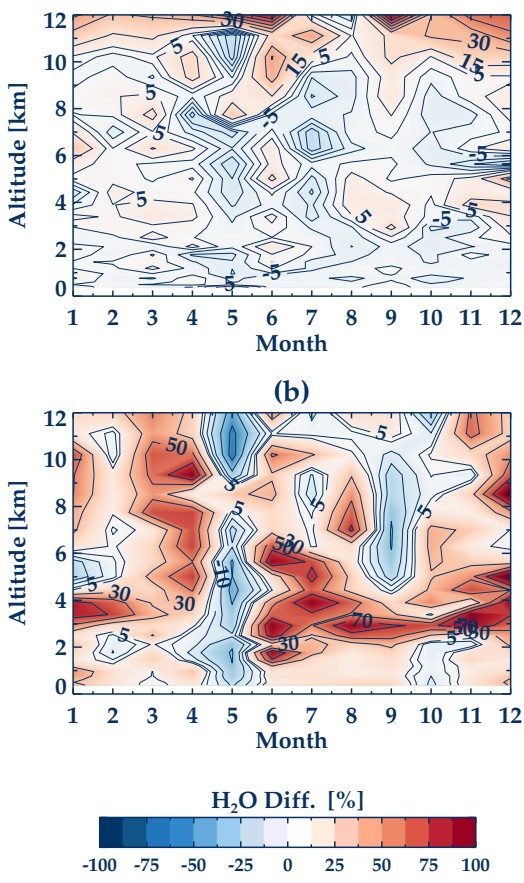

**Figure 2.** (a) Multi-annual and monthly mean percentage differences between radiosonde and FPH $H_2O$ measurements. (b) Multi-annual and monthly-mean percentage differences between NIWA-UKCA output and FPH $H_2O$.

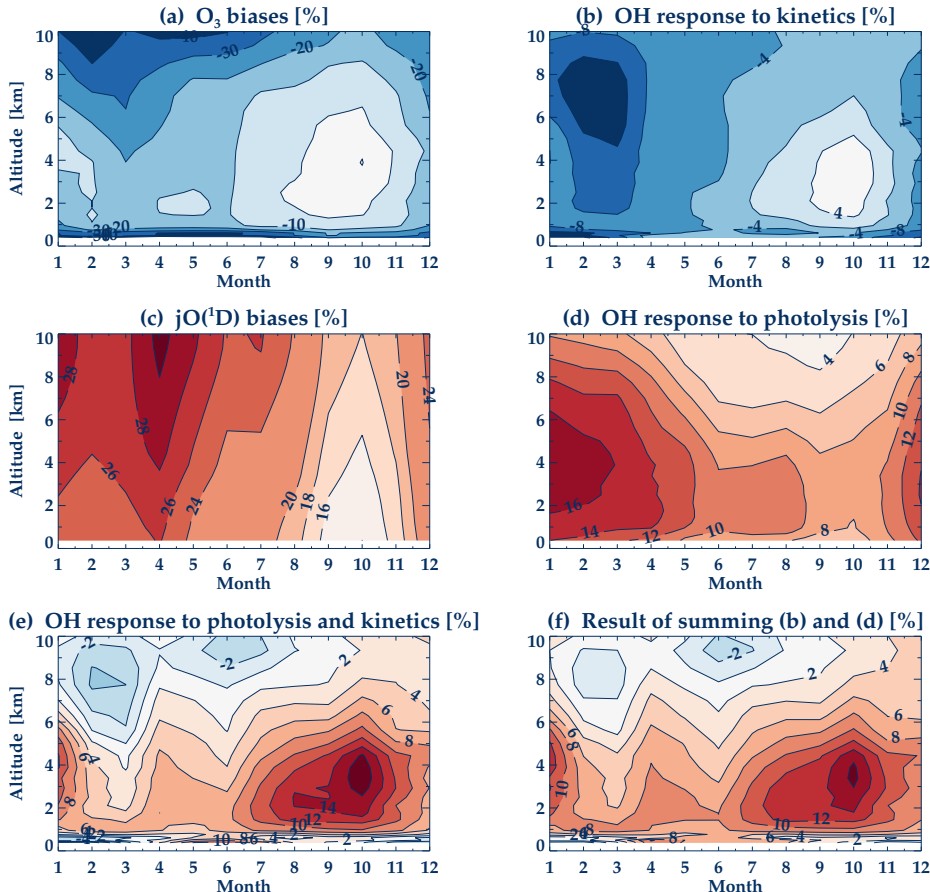

**Figure 3.** Multi-annual and monthly-mean OH responses to $O_3$ biases between observations and the reference simulation. (a) Difference in $O_3$ (%) between ozone sonde and NIWA-UKCA ozone, relative to NIWA-UKCA ozone as prescribed in the reference simulation. (b) OH difference (%) relative to the reference simulation accounting only for the kinetics effects of $O_3$ differences (e.g. with $jO(^1D)$ unchanged). (c) Difference in $jO(^1D)$) (%) relative to the reference simulation. (d) OH difference (%) relative to the reference simulation accounting only for $jO(^1D)$ differences (e.g. with $O_3$ unchanged). (e) OH differences relative to the reference simulation considering the combined kinetics and photolysis effects. (f) Sum of (b) and (d).

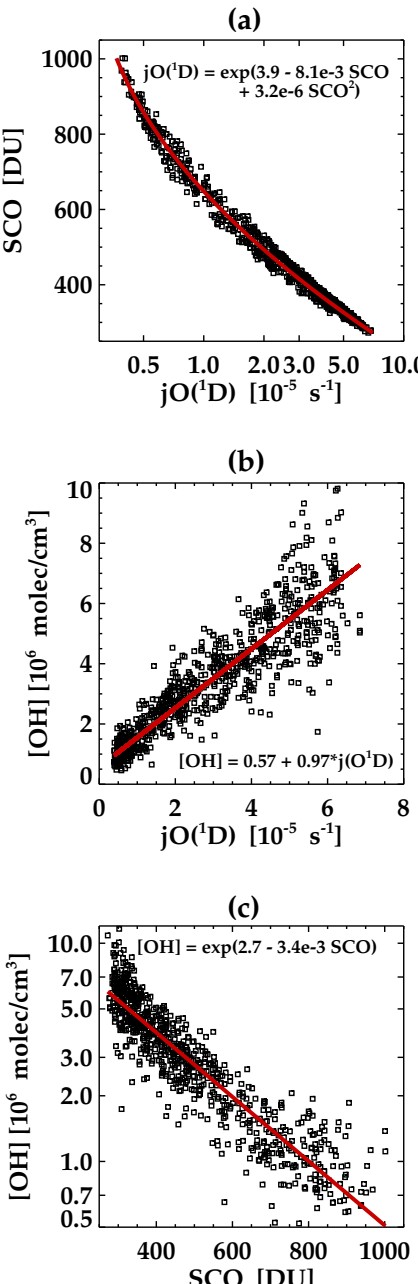

**Figure 4.** (a) Scatter plot of $j_{O1D}$ with the slant column of $O_3$ (SCO) at 6 km of altitude. (b) Same, but for $j_{O1D}$ and OH. (c) Same, but for the SCO and OH. The results shown in this figure are those obtained from the combined simulation (kinetics and photolysis effects). Red lines denote least-squares fits between the variable pairs. The best fits are stated in the panels, with [OH] in units of $10^6$ molec/cm$^3$, $j_{O1D}$ in units of $10^{-5}$ s$^{-1}$, and the SCO in Dobson Units.

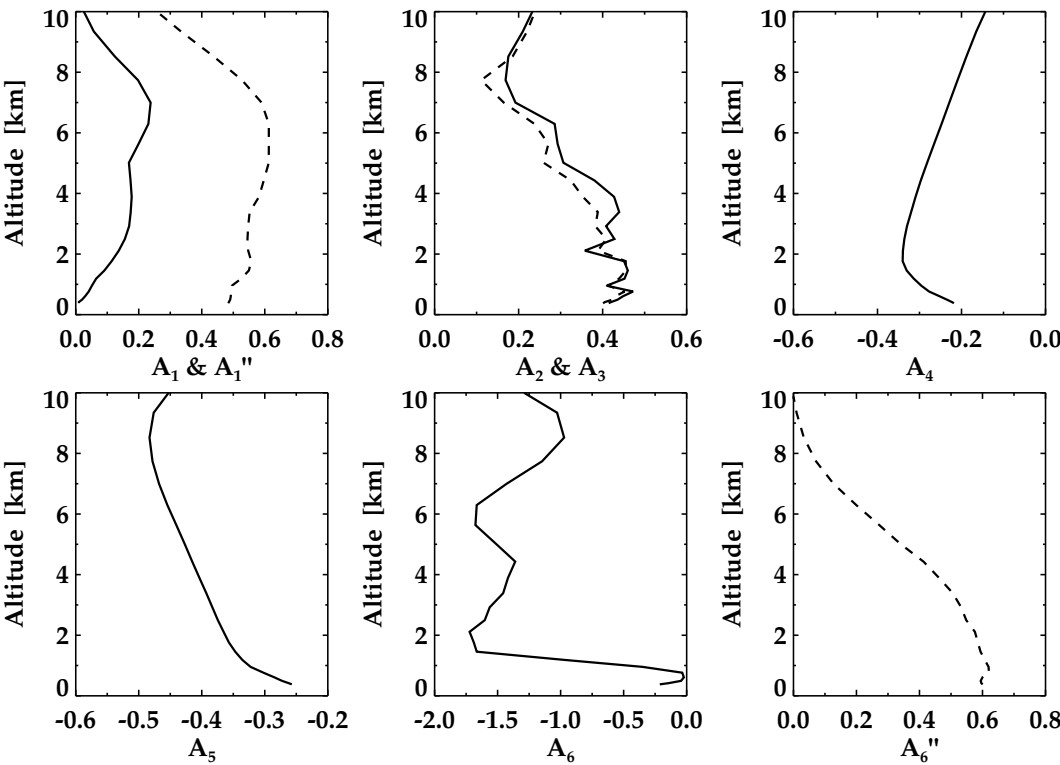

**Figure 5.** Sensitivity coefficients 'A$_i$' between OH and each perturbation variable: In the calculation, multi–annual mean relative differences in OH and in the forcing are ratioed. (a) Sensitivity of OH to changes in O$_3$ levels (kinetics effect) denoted by A$_1$ (solid line) and to changes in $j$O($^1$D) due to changes in O$_3$ (photolysis effect) denoted by A$_1$" (dashed line); (b) sensitivity of OH to changes in radiosonde – NIWA-UKCA CCM H$_2$O (A$_2$ solid line) and to changes in ERAI – NIWA-UKCA H$_2$O (A$_3$ dashed line); (c) sensitivity of OH to changes in CH$_4$ (A$_4$); (d) sensitivity of OH to changes in CO (A$_5$); (e) sensitivity of OH to changes in temperature (kinetics effect) denoted by A$_6$; (f) sensitivity of OH to changes in $j$O($^1$D) due to changes in temperature (photolysis effect) denoted by A$_6$".

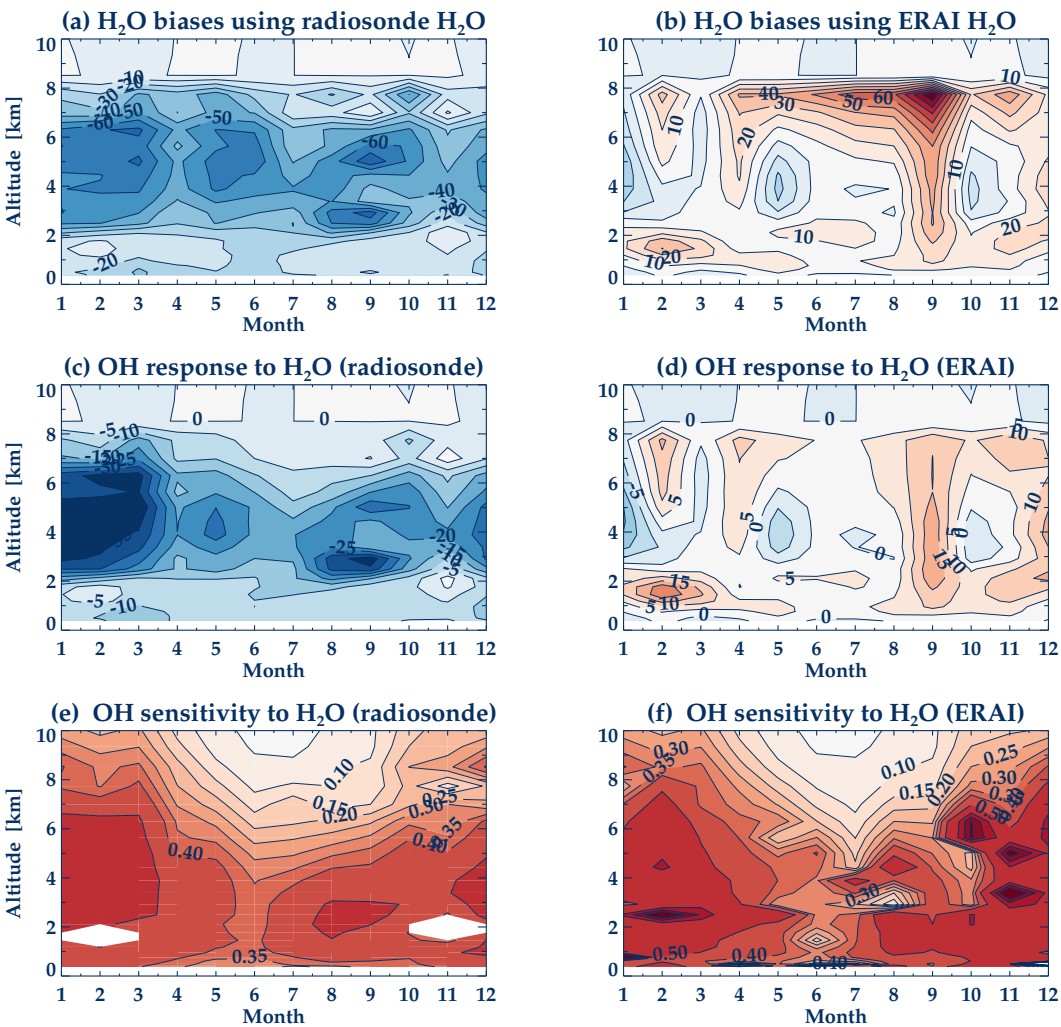

**Figure 6.** Multi–annual and monthly–mean OH responses to $H_2O$ between perturbation simulations and the reference simulation. (a) Radiosonde − NIWA-UKCA $H_2O$ (%) relative to the reference simulation. (b) ERAI − NIWA-UKCA $H_2O$ (%) relative to the reference simulation. (c) OH difference (%) relative to the reference simulation between simulations using radiosonde and NIWA-UKCA $H_2O$ (panel a). (d) OH differences (%) relative to the reference simulation between simulations using ERAI and NIWA-UKCA CCM $H_2O$ (panel b). (e) Ratio of relative OH changes (panel c) to relative changes in $H_2O$ (panel a). (f) Ratio of relative OH changes (panel d) to changes in $H_2O$ (panel b). Above 8 km NIWA-UKCA $H_2O$ was used in both cases. Therefore, differences with respect to the reference simulation are close to 0.

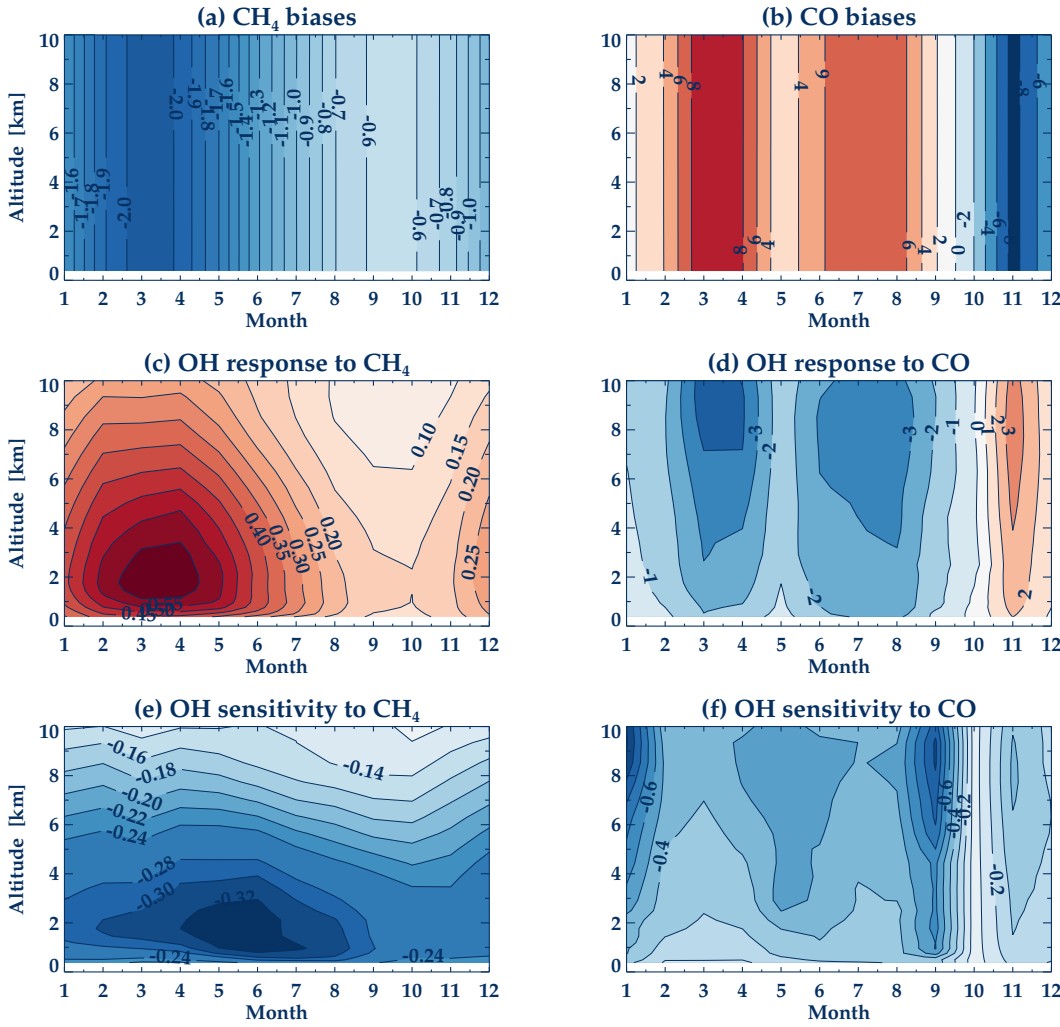

**Figure 7.** Multi–annual and monthly–mean OH responses to CH₄ and CO biases between observations and the reference simulation. (a) Difference in CH₄ (%) relative to the reference simulation. (b) Difference in CO (%) relative to the reference simulation. (c) OH difference (%) relative to the reference simulation caused by the CH₄ change (panel a). (d) OH difference (%) relative to the reference simulation caused by the CO change (panel b). (d) Ratio of relative OH changes (panel c) to relative changes in CH₄ (panel a). (f) Ratio of relative OH changes (panel d) to relative changes in CO (panel b).

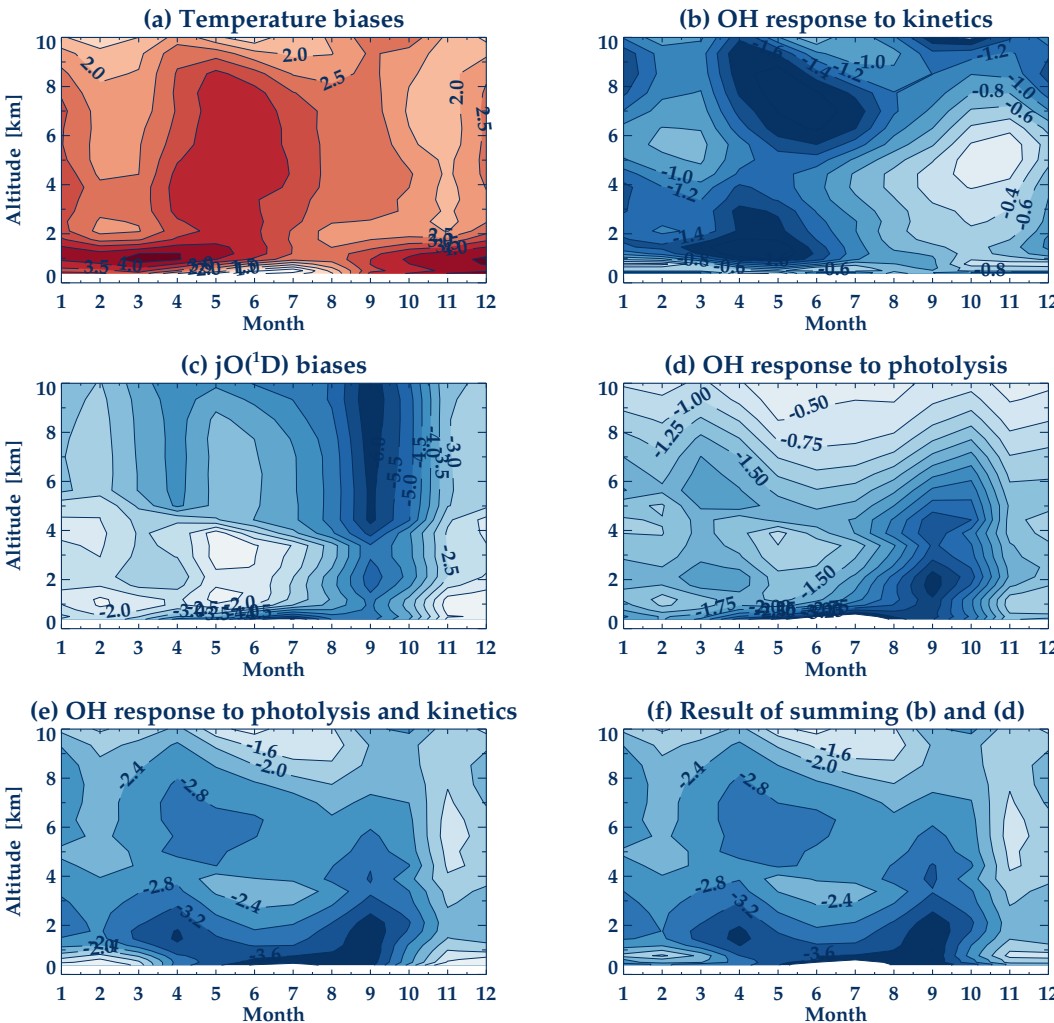

**Figure 8.** Multiannual and monthly-mean OH responses to temperature biases between observations (radiosonde and NCEP/NCAR temperature) and the reference simulation. (a) Difference in radiosonde and NCEP/NCAR temperature (K) relative to the reference temperature. (b) OH difference (%) relative to the reference simulation accounting only for the kinetics effects of temperature differences (e.g. with $j\mathrm{O}(^1\mathrm{D})$ unchanged). (c) Difference in $j\mathrm{O}(^1\mathrm{D})$ (%) relative to the reference simulation. (d) OH difference (%) relative to the reference simulation accounting only for $j\mathrm{O}(^1\mathrm{D})$ differences (i.e. with temperature unchanged). (e) OH differences relative to the reference simulation considering the combined kinetics and photolysis effects. (f) Sum of (b) and (d).

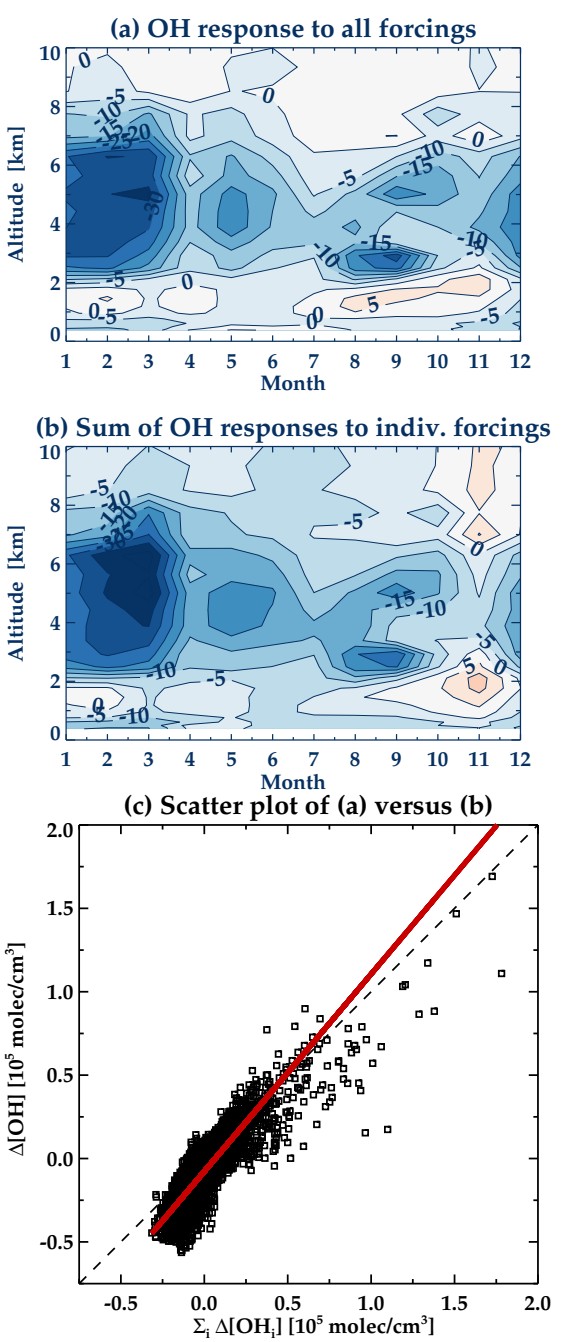

**Figure 9.** (a) Multi–annual and monthly–mean percentage difference in OH between a simulation with bias–correction applied to all five fields and the reference simulation. Radiosonde $H_2O$ is assumed below 8 km. (b) Summation of all the single forcing contributions as expressed by the right hand side of Eq. (4). Radiosonde $H_2O$ is assumed below 8 km. (c) Scatter plot of the response of OH to the combination of all forcings (vertical axis, denotes as $\Delta[OH]$) versus the summation of the OH response to individual forcings (horizontal axis) as expressed by the right hand side of Eq. (4) (denoted by $\sum_i \Delta[OH_i]$). The red solid line denotes an orthogonal fit. The black dashed line is the diagonal.

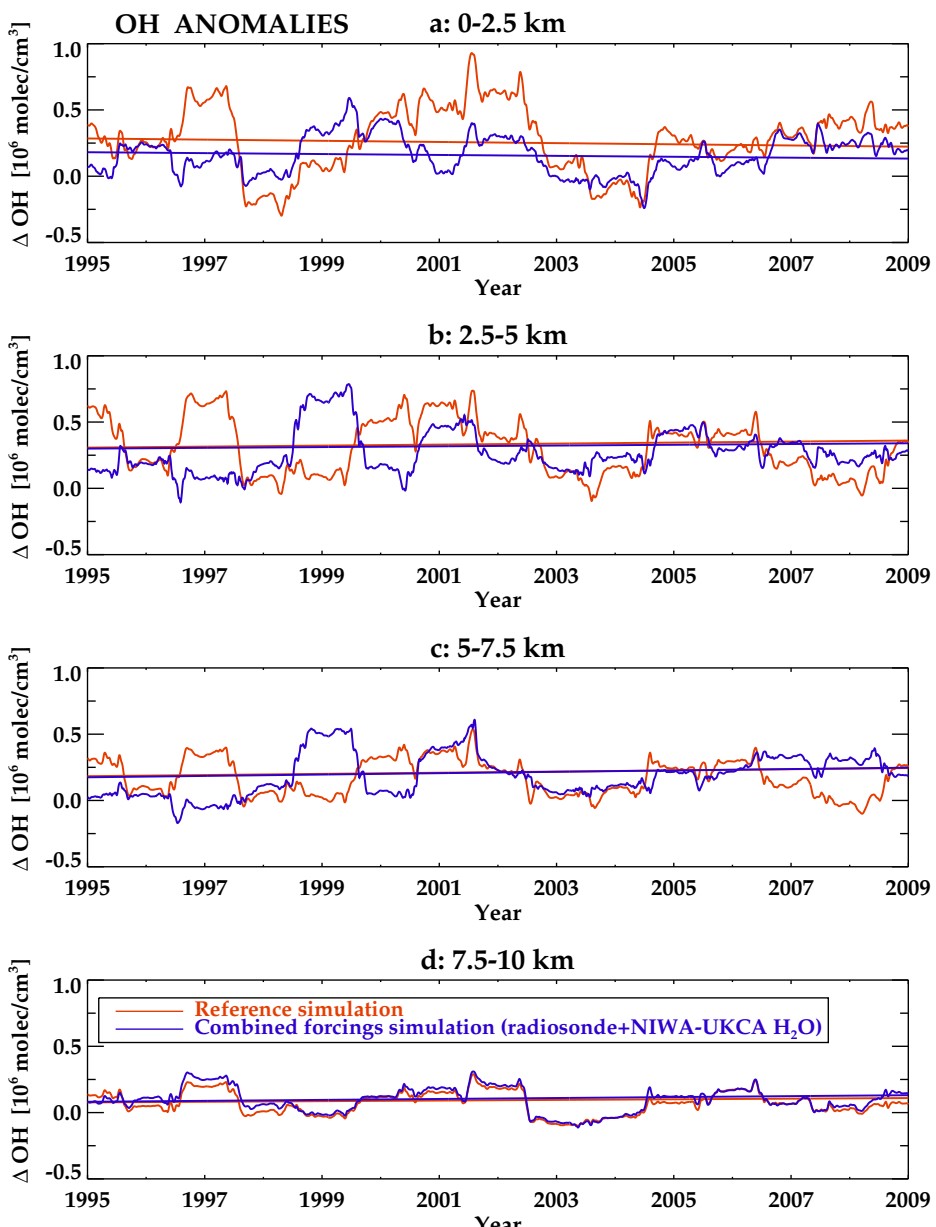

**Figure 10.** Variability and trends of the OH anomalies at different altitudes: (a) 0-2.5 km, (b) 2.5-5 km, (c) 5-7.5 km, and (d) 7.5-10 km. The red solid line is the time series of the reference simulation and the blue solid line is the combined forcings simulation considering radiosonde – NIWA-UKCA $H_2O$.

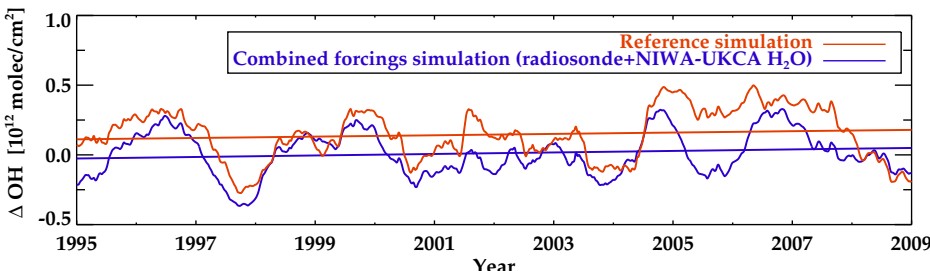

**Figure 11.** Variability and trend of the OH column anomaly. The red solid line is the time series of the reference simulation and the blue solid line is the combined forcings simulation considering radiosonde – NIWA-UKCA H$_2$O.

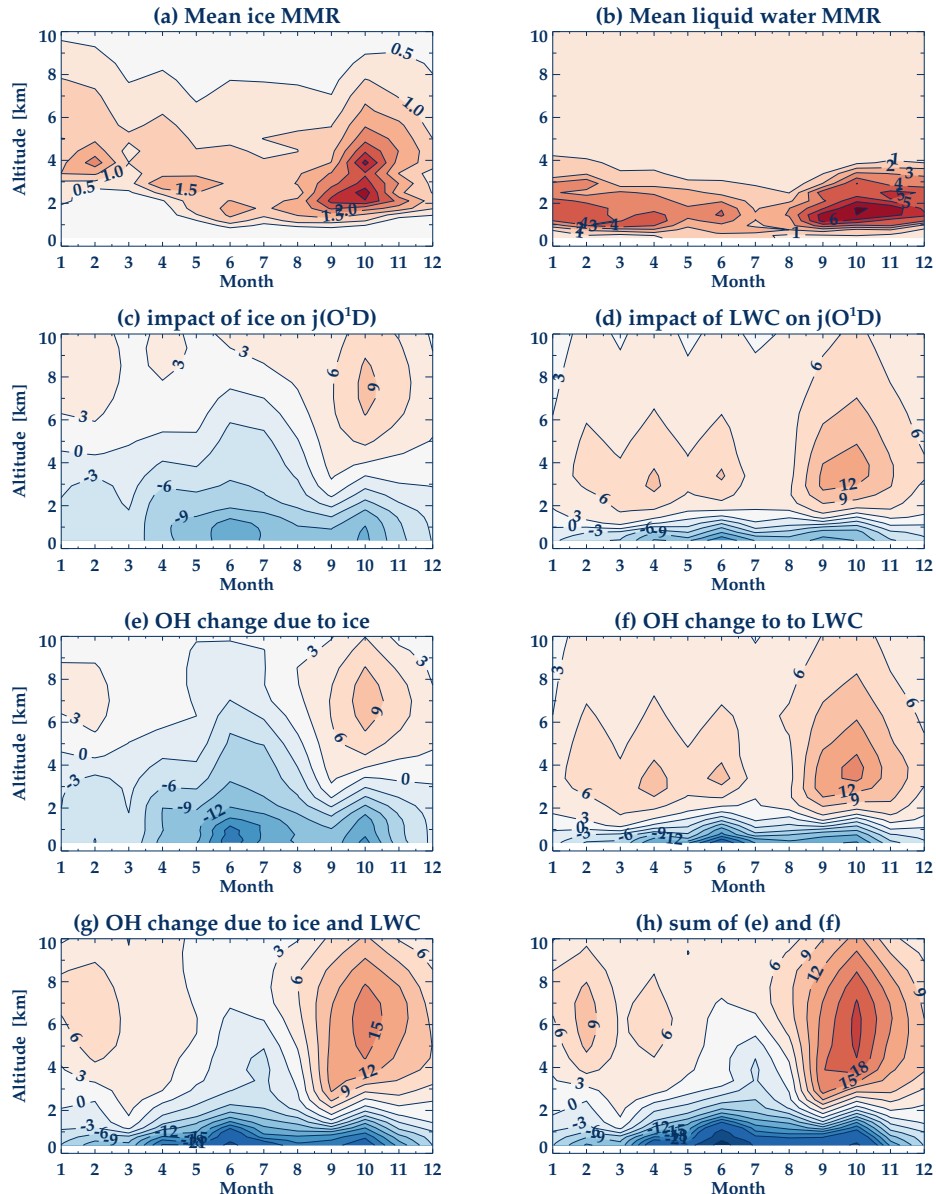

**Figure 12.** Multi–annual and monthly–mean OH responses to the presence of clouds. Multi–annual and monthly mean (a) ice content ($10^{-5}$ kg/kg). (b) liquid water content ($10^{-5}$ kg/kg). (c) Response of $j_{O1D}$ (%) to the presence of ICs relative to the cloud–free reference simulation. (d) response of $j_{O1D}$ (%) to the presence of LWCs relative to the cloud–free reference simulation. (e) response of OH (%) to the presence of ICs relative to the cloud–free reference simulation. (f) Response of OH (%) to the presence of LWCs relative to the cloud–free reference simulation. (g) response of OH (%) to the presence of both LWCs and ICs relative to the cloud–free reference simulation. (h) Sum of (e) and (f).