# Peer review of "Assessing the sensitivity of the hydroxyl radical to model biases in composition and temperature using a single-column photochemical model for Lauder, New Zealand"

_Atmospheric Chemistry and Physics, 2016_

## Referee Comment (RC1) · Anonymous Referee #1 · 15 Jul 2016

Assessing the sensitivity of the hydroxyl radical to model biases in composition and temperature using a single-column photochemical model for Lauder, New Zealand

López-Comí et al., Atmos. Chem. Phys. Discuss., doi:10.5194/acp-2016-448, 2016

The authors report a study of the effects on OH of constraining model calculations to observations of ozone photolysis rates and concentrations of ozone, water vapour, CO and methane, as opposed to model derived fields for these variables. The model used is a single column photochemical model, based on the NIWA-UKCA model, and enables a focus on the model chemistry owing to removal of transport and physical

processes whilst also demonstrating changes in chemical effects on OH as a function of altitude.

My major comment with the paper regards a lack of detail, and insufficient attention given to the wider applicability of the results obtained in Lauder, New Zealand, to global chemistry-climate modelling.

My major comments are:

Abstract: The general trends for changes in OH are described but these should be quantified throughout.

Introduction: The introduction is rather short and lacking in detail. The rationale for studying OH is brief, and the paper would benefit from an expanded discussion of why it is such an important model target. The statement that 'considerable disagreement among . . . models' should be quantified, and given that the abstract describes the possibility of this work explaining 'differences in simulated OH between global chemistry models and relative to observations' some discussion of relevant previous studies is warranted. Differences in model outputs observed in intercomparisons such as AC-CMIP could be of interest here, and would help place this paper and its results in greater context of previous work.

The Emmerson et al. papers referenced (line 47) refer to box models, some reference to single-column models, and examples of their use, should be given. There is no reference given for Lauder being 'known for its clean air' (line 49), or much detail given the 'large diversity of available measurements' (line 50). Apart from O3, H2O, CO and CHÂň4, what species are measured? Are there measurements of NOx (what are the average values?) or other VOCs?

Line 173: Please clarify that the changes in modelled O3 (Fig. 2a) are a result of constraining to the observations and not a model result. Is there any explanation for the increases in spring and decreases in autumn compared to the reference simulation?

Or for the altitude dependence?

Line 181: Is the 5 % increase an average value over all altitudes/seasons? Please clarify.

Line 185: The statement that the increases in OH are the result of increases in jO1D seems rather obvious given that this is the only parameter that has been changed.

Line 188: Please explain (and discuss) more clearly what you mean by the statement that the magnitudes of the kinetics and photolysis effects are comparable. Figures 2c and 2d show the changes to jO1D and OH respectively, how do these suggest anything about the O3 bias? The values shown in Figures 2a and 2b, which do correspond to the kinetics effects, are not comparable or similar to those in Figures 2c and 2d.

Line 193: What is the significance of a near exponential relationship? Does it have a physical basis? From the plot it is not clear that there is a near exponential relationship, if there is and it is significant, please show it on the plot and give the parameters describing the relationship. Does Figure 3 show data from all altitudes? The discussion comments on an altitude of 6 km, how does this relate to the data shown in the figure?

Line 199: Again, explain the significance of the exponential relationship and give the parameters describing it.

Line 213: The percentages given in the discussion are given as fractions in the figures, please change one or the other for consistency.

Line 235: What is the fraction of the total OH loss to CH4 and CO in the model? It is not clear from the discussion what fraction of the total OH loss occurs due to reactions with CH4 and CO, what are the implications of the presence of other species, and thus the applicability of the results obtained in this work to more polluted regions. The OH concentrations shown in Figure 3 seem particularly high.

If the CH4 observations are different from the reference simulation by only ∼2 % please explain the reported 40 % sensitivity of OH to the change in CH4. The discussion

refers to the percentage changes in OH shown in Figure 6e/6f, but these do not show percentage changes. The discussion should be consistent with the figures in terms of the way the differences are expressed. Please provide some discussion of the use of d ln(OH)/ d ln(CH4) (or CO) in Figure 6.

Line 275: Is OH + CH4 the dominant OH sink in the model? What is the change in the kinetics of the reaction for the temperature change applied to the model?

Line 310: What is the significance of this equation? Can it be applied to other models? Can values for the parameters be tabulated for various altitudes (or can altitude-dependent parameters be given?). How valid is the assumption that the OH response is linear to changes in the forcings? As stated, Figure 8c suggests this is not a valid assumption.

Line 385: Please give some examples (and references!) of underestimated CH4 lifetimes by NIWA-UKCA and comparisons with other accepted estimates. An expanded introduction will help with this.

Minor comments: Line 11: 'Its impact...', please change this to 'The impact of O3 ... ' for clarity.

Line 32: 'in-situ' to 'in situ'.

Line 60: Please spell out NIWA in full.

Line 71/line 135: What determines the concentrations of these species in the model if there are no emissions? Are they constrained to observations? Set to zero?

Page 101: 'Vertically integrated ozone produced here' – please reword, do you mean 'produced in this way'.

Line 161: Please replace 'a' and 'b' with 'k' in keeping with convention, and label the different 'k' appropriately to distinguish between reactions (i.e. ka, kb or k1, k2).

Line 290: Space in '5K'.

Line 329: 'sky' to 'skies'.

Line 336: Please change the word 'combinedly'.

Line 373: Please change 'chemical equilibrium' to 'chemical steady state'.

Figure 1: Panel e, please remove the degree symbol.

Figure 2: Panel f, presumably this should refer to panels 2b and 2d?

Figure 3: Please remove the titles to the plots and leave just the labels a, b and c. See comments above regarding the exponential relationships - please give the parameters (and fit statistics) for the relationships described if these are important. If they are, why mention them?

Figure 4: The data shown in the plots are given as percentages in the discussion. Please see comments above regarding consistency.

Figure 5: Please clarify in the caption that panels e and f refer to plots a&b and c&d, respectively. The analysis d ln(OH) / d ln(H2O) is not explicitly referred to in the text (likewise for Figure 6).

Figure 6: Figure 6e in the caption is referred to as Figure 6d.

Figure 8: Panel c, please explain the significance of the dashed and red lines.

---

## Referee Comment (RC2) · Anonymous Referee #2 · 23 Aug 2016

Manuscript Number: acp-2016-448 Title: Assessing the sensitivity of the hydroxyl radical to model biases in composition and temperature using a single-column photochemical model for Lauder, New Zealand Authors: L. López-Comí et al.

General Comments: This paper shows the influence of biases in modeled $O_3$, $H_2O$, CO, $CH_4$, and temperature on modeled OH as investigated using a single-column model and observations over Lauder, New Zealand. Model fields of the parameters listed above are replaced with observations, and the photochemical single-column model is used to re-calculate OH and establish changes and sensitivities in OH rel-

ative to a reference run. Impacts of O3 and temperature biases are further examined by separating kinetic and photolytic effects. Long-term OH trends and effects of clouds on OH are briefly examined.

While this analysis is somewhat limited in scope and some aspects of the discussion are quite cursory, tropospheric OH is an important issue requiring varied and novel approaches to build on the community's understanding. With some revisions, this paper would contribute a useful method to help identify how model representation of OH can be improved and why model versus empirical estimates of the CH4 lifetime differ.

Specific Comments:

Line 10: Please provide some quantification for these results. Particularly useful would be an indication of how much H2O differed between the model and observations as well as a quantification of how OH changed in response. The same could be done for subsequent species.

Line 47: You state two paragraphs above that "in-situ measurements of OH do not sufficiently constrain its global abundance." Here, you cite two Emmerson papers that do exactly that as justification for your SCM approach. I understand that constraining OH globally is not your aim, but the two statements still seem contradictory. It would be worthwhile to strengthen your justification for this analysis - what questions are you seeking to answer? What role can this approach play in constraining global OH, even if there are limitations?

Line 75: The number of species and reactions represented in the NIWA-UKCA chemical mechanism seems low, at least compared to explicit schemes like the MCM (easily into the hundreds of species and thousands of reactions). Might be worth noting why it's important to maintain consistency with the NIWA-UKCA model/why you wouldn't want a more detailed mechanism in your SCM, since "assessing fast photochemistry" is your goal.

Line 326: This section should be expanded. Even though the trends are not significant, they can still be quantified, and numbers here compared to values in the other studies you cite. Also, trends shown, for example, in Montzka et al., 2011 are derived from a globally, vertically integrated [OH] calculation, so separation into altitude bins, while useful, may not be the best comparison. I realize you don't seek to look at global [OH], but at least for this location, you could include a vertically integrated OH trend to compare to Montzka et al. In addition to quantifying the trend, you could also quantify the interannual variability.

Line 430: The reader is likely interested to hear your hypotheses on why NIWA-UKCA is too moist and O3 is too high, even if further investigation is beyond the scope of this paper.

Minor Comments:

Line 11: Reference to O3's kinetics and photolysis effects is unclear until defined in the body of the paper; please rephrase for abstract. Assertion that both are of similar magnitude does not seem well-supported, as pointed out above

Line 12: Sentence about OH being inversely related to CO and CH4 is unnecessary for an audience familiar with OH.

Line 19: Use of "less-than-additive" is vague, especially for an abstract. Instead of focusing on how the LWC and IC effects combine, it would probably be more informative to note the quantitative results of the combined LIC simulation, if that is the more realistic one. This would likely be of greater interest to the reader.

Line 20: Please quantify trends as well, even if they are insignificant.

Line 101: use of word "produced" is unclear

Line 115: You make the case for not trusting radiosonde H2O data above 8 km, but how about the NIWA-UKCA output? Does modeled H2O agree well with FPH? A figure addressing this point might be suited for supplemental material.

Line 131: It would be helpful to address some anomalous behavior in the H2O profiles shown in Fig. 1: in the winter (presumably) of 1996, and to a lesser extent in other years, there are sudden high temperatures around 40-60 km - what's the cause of this? Is there evidence of this truly happening in the atmosphere or is it a result of interpolation, instrument artifact, etc?

Line 137: What are the native temporal and spatial resolutions of this simulation? Do you also interpolate spatially?

Line 174: Make clear that you're discussing local O3, or the "kinetics" effect

Line 177: The sentence "The largest impact is in the free troposphere where these differences vary with altitude." is a bit vague. Please be specific; what are the differences you're referring to, and how do they vary?

Line 188: I'm not sure what you mean by the statement that kinetics and photolysis effects of the O3 bias are comparable. Based on my interpretation of the contours in Fig. 2, the response of OH to kinetic effects is both positive and negative, depending on the month and ranges from -12 to +4%; the response to photolysis effects is only positive, about 4-16%. The two effects somewhat cancel around Feb-June - is this what you're referring to? Please clarify.

Line 241: The statistic that sensitivity of OH to CH4 changes peaks at ∼40% can be easily misinterpreted as the OH response; it may be helpful to highlight both the max OH response as well as the OH sensitivity to avoid confusion.

Line 295: You stated above that the O'Connor et al. result may be due to cancellation of positive and negative temperature biases, but you show that temperatures at Lauder are cold-biased, so saying that your result of small impact of temp on OH corroborates that of O'Connor et al. seems like an apples-to-oranges comparison. I'd suggest reframing the discussion of O'Connor et al. – the small impact on OH in O'Connor et al. could have been due either to cancellation of temp biases or to low sensitivity of OH to

temp changes, and your result suggests the latter? Or something to that effect.

Line 320: Care to hypothesize about what might be causing these non-linearities?

Line 366: what do you mean by "slow chemistry"? My best guess is something like oxidation of CH4 (long-lived), yet that is considered here, so I'm not sure your intended meaning.

Line 382: Again, would like to see quantification here; how much is the H2O overestimated?

Line 387: Please include some references, particularly when citing "accepted literature estimates"

Line 400: "...small reduction ...due to the strong dependence of OH+CH4 on temperature." This does not logically follow; you'd think, with a strong dependence, that you should see a large reduction. Please clarify.

Line 406: Thank you for quantifying the H2O bias! I think this statistic would be better suited to earlier paragraph on H2O, plus repeat in Section 3 and in abstract.

Fig. 2: The use of both blue and red for strictly positive values is slightly confounding at first glance (panels (c)-(f)); if possible, would help to keep the white contour at value 0 (applies to various upcoming figures as well).

Table 1: Is the O3 photolysis effect analysis done in an altitude-dependent manner? I.e., is a new j(O1D) value calculated at each vertical point based on an overhead O3 column that's adjusted to account for the strat column plus the partial tropospheric column overhead? I did not see any details regarding this in the text.

Fig. 5: Use of d(ln(OH))/d(ln(H2O)) is not mentioned in text, is not consistent with "Ai" terms in Fig. 4; please either justify switching metrics or maintain consistency (same with Fig. 6).

Fig. 9: y-axis label is misleading since, based on the caption, this shows OH anomalies. Perhaps include word "Anomaly" or a "delta" sign. Also, the values chosen for the y-axis tick marks are not easy to work with; it would be nice if they were adjusted to lie along round calculation-friendly values (e.g. increments of 0.5 instead of 0.417 in panel c). Also, how are these anomalies calculated, relative to what?

Technical Corrections:

Line 23: Use of word atmospher-e/-ic 3x

Line 29: "plays a important" should be "plays an important"

Line 50: "long time series" wording seems off; perhaps "long record of observation" instead. I'm also curious at this point, how long is long? Perhaps give an earliest year of observation.

Line 53: I think "Section 1" should be "Section 2"?

Line 250: "altitide" should be "altitude"

Line 256: spelling of the word "assess" is incorrect

Line 275: use of "explicitly" does not add meaning to this sentence but makes it read awkwardly; I suggest removing

Line 279: "nearly completely linearly" should be "nearly linearly"

Line 324: instead of "altitude bands", I more often see the phrase "altitude bins"

Line 378: use of word "effect" doesn't seem quite right; it's the bias you're correcting

---

## Author Response (AR1)

Dear Editor,

In response to the comments made by the reviewers, we have substantially improved the presentation of the paper. In particular, we have made the following changes to the manuscript:

- 1. The abstract now states some key numerical results (previously it had been worded in a mostly qualitative way).
- 2. The introduction is now much expanded, containing a more exhaustive discussion of the literature.
- 3. We corrected a minor error with the temperature climatology: Data above 25 km were actually not MOPI measurements, they were reanalyses and other data as used for retrieval of ozone from MOPI measurements.
- 4. The description of the model has been expanded, addressing several of the reviewers' comments.
- 5. We now state regression fits to every pair of the SCO, j(O1D), and OH variables displayed in fig. 4.
- 6. We have expanded the Conclusions section, now charting a way forward how this study could be of use in the future to address problems with simulating OH in global models.
- 7. The figures have been overhauled. We now consistently use the blue-red scale, with white reserved for the 0 value (if present).

We hope the paper is now suitable for publication in ACP, and are looking forward to hearing from you.

Best regards,

0 mgt

Olaf Morgenstern (on behalf of all co-authors).

Atmos. Chem. Phys. Discuss., doi:10.5194/acp-2016-448-AC1, 2016 © Author(s) 2016. CC-BY 3.0 License.

ACPD
We thanks the reviewer for his/her thoughtful comments. The reviewers comments are repeated below in italics.

The authors report a study of the effects on OH of constraining model calculations to observations of ozone photolysis rates and concentrations of ozone, water vapour, CO and methane, as opposed to model derived fields for these variables. The model used is a single column photochemical model, based on the NIWA-UKCA model, and enables a focus on the model chemistry owing to removal of transport and physical

processes whilst also demonstrating changes in chemical effects on OH as a function of altitude.

My major comment with the paper regards a lack of detail, and insufficient attention given to the wider applicability of the results obtained in Lauder, New Zealand, to global chemistry-climate modelling.

We are now providing significantly more detail, improving the presentation of the results, and have also expanded the conclusions section in response to the reviewer's concern about insufficient coverage of the wider significance of the research.

1. Abstract: The general trends for changes in OH are described but these should be quantified throughout.

We have modified the abstract trying to strike a balance between giving some quantitative information and not overloading it with numbers which can make the abstract unreadable. This was also in response to a comment by the 2nd reviewer.

2. Introduction: The introduction is rather short and lacking in detail. The rationale for studying OH is brief, and the paper would benefit from an expanded discussion of why it is such an important model target. The statement that 'considerable disagreement among . . . models' should be quantified, and given that the abstract describes the possibility of this work explaining 'differences in simulated OH between global chemistry models and relative to observations' some discussion of relevant previous studies is warranted. Differences in model outputs observed in intercomparisons such as ACCMIP could be of interest here, and would help place this paper and its results in greater context of previous work.

We have now considerably expanded the introduction, giving quantitative information on the disagreement e.g. found in ACCMIP models. We hope that this addresses the reviewer's concern. Interactive comment

3. The Emmerson et al. papers referenced (line 47) refer to box models, some reference to single-column models, and examples of their use, should be given. There is no reference given for Lauder being 'known for its clean air' (line 49), or much detail given the 'large diversity of available measurements' (line 50). Apart from O3, H2O, CO, and CH4, what species are measured? Are there measurements of NOx (what are the average values?) or other VOCs?

References to single-column models have been added in the text, as well as their different applications. Single-column models for OH chemistry, other than our own, to our understanding do not exist. The paper now states that the Emmerson et al. papers refer to the development and application of box models, rather than SCMs. See paragraph 4 of the introduction.

A reference attributing Lauder to be a clean air site has been added.

In addition to the measurements used here, Lauder produces measurements of total-column NO2, total-column BrO, and FTS measurements of a variety of species (mostly as total columns) that are reported at NDACC. For this study, relatively high-resolution profile information is needed which in the troposphere is only achieved by the ozone sondes (which produce O3, H2O, and *T* profiles). Near-surface NO2 has only been measured episodically at Lauder; its abundance has generally been at or near the detection limit. We don't know what the averages would be. These measurements are unsuitable to constrain the SCM with. Lauder is well known for its total-column NO2 measurements but these are dominated by a stratospheric contribution and cannot be used to infer tropospheric abundances of NO2.

A few VOCs have been derived as total-columns from FTS measurements ( $C_2H_6$ , HCHO, HCN, CH3OH). These species are all orders of magnitude less abundant than CH4, and biases in them would have a small direct impact on OH. For HCHO, there is a considerable discrepancy between modelled and measured total columns, which we cannot fully explain (Zeng et al., 2015) and which may

ACPD
be indicative of a knowledge gap regarding VOCs at Lauder. A more exhaustive discussion of this is however beyond the scope of this paper.

4. Line 173: Please clarify that the changes in modelled O3 (Fig. 2a) are a result of constraining to the observations and not a model result. Is there any explanation for the increases in spring and decreases in autumn compared to the reference simulation? Or for the altitude dependence?

In subsection 3.1 (OH sensitivity to  $O_3$  biases), paragraph 2, we now clarify that the  $O_3$  difference (fig. 2a) is indeed NIWA-UKCA ozone versus ozonesonde data. The caption of fig. 2a is also changed accordingly.

Exploring the causes of the ozone and temperature biases in NIWA-UKCA is the subject of an ongoing investigation; we don't fully understand why these biases occur either. Exploring these causes is beyond the scope of this paper; here we focus only on the consequences of these biases for OH.

5. Line 181: Is the 5% increase an average value over all altitudes/seasons? Please clarify.

We now clarify that this pertains only to the spring season and the altitude range of 2-6 km. 5% is about the maximum difference.

6. Line 185: The statement that the increases in OH are the result of increases in  $j_{O1D}$  seems rather obvious given that this is the only parameter that has been changed.

We have rephrased this sentence. Indeed there is no surprise here, but we have gained a simple quantification of the impact of TCO errors on OH.

7. Line 188: Please explain (and discuss) more clearly what you mean by the statement that the magnitudes of the kinetics and photolysis effects are comparable. Figures 2c and 2d show the changes to  $j_{O1D}$  and OH respectively, how do these Interactive comment

suggest anything about the  $O_3$  bias? The values shown in Figures 2a and 2b, which do correspond to the kinetics effects, are not comparable or similar to those in Figures 2c and 2d.

The text now makes this clearer. Indeed the patterns are different, but the ranges of values are comparable.

8. Line 193: What is the significance of a near exponential relationship? Does it have a physical basis? From the plot it is not clear that there is a near exponential relationship, if there is and it is significant, please show it on the plot and give the parameters describing the relationship. Does Figure 3 show data from all altitudes? The discussion comments on an altitude of 6 km, how does this relate to the data shown in the figure?

The near-exponential is motivated by the Lambert-Beer Law, which links attenuation to optical thickness. The finding is still semi-empirical because the solar UV light is not monochromatic and the optical thickness of ozone is wavelength dependent. The plot only shows how OH would respond to a systematic change of the SCO. We now provide empirical fits in all three panels. In the case of SCO versus  $j_{O1D}$ , we account for the curvature by fitting a parabola to log  $j_{O1D}$ . The fit parameters are stated in the panels, and some text is added to this effect.

9. Line 199: Again, explain the significance of the exponential relationship and give the parameters describing it.

The same as above.

10. Line 213: The percentages given in the discussion are given as fractions in the figures, please change one or the other for consistency.

We have changed the text to make it consistent with the panels.

11. Line 235: What is the fraction of the total OH loss to  $CH_4$  and CO in the model? It is not clear from the discussion what fraction of the total OH loss occurs due
to reactions with  $CH_4$  and CO, what are the implications of the presence of other species, and thus the applicability of the results obtained in this work to more polluted regions. The OH concentrations shown in Figure 3 seem particularly high.

Unfortunately we cannot straightforwardly diagnose in the SCM what the fraction of OH lost to CH4 or CO is. For NIWA-UKCA, such diagnoses exist for the global domain. In the troposphere, OH + CO is the leading sink process, but this reaction does not change HOx, and OH is well buffered w.r.t. changes in CO. For HOx, the dominant sink processes are self-reactions of HOx (HO2 + HO2  $\rightarrow$  H2O2, HO2 + OH  $\rightarrow$ H2O) competing with OH + CH4. However, the methane impact is complicated by the oxidation products. In short: a satisfying answer to this question would require a much more comprehensive investigation.

Regarding the high values of OH in figure 3, again this is for the 6 km level (where OH concentrations are larger than at the surface). We have made this explicit in the figure caption.

12. If the CH4 observations are different from the reference simulation by only  $\sim 2\%$  please explain the reported 40% sensitivity of OH to the change in CH4. The discussion refers to the percentage changes in OH shown in Figure 6e/6f, but these do not show percentage changes. The discussion should be consistent with the figures in terms of the way the differences are expressed. Please provide some discussion of the use of d ln(OH)/ d ln(CH4) (or CO) in Figure 6.

The number of 0.4 was an error. OH increases by > 0.6% when CH4 is increased by 2% in March (figure 6c). The relative sensitivity  $A_{CH4} = \partial \ln OH / \partial \ln CH_4$ is no longer expressed in percent. It maximizes, in absolute terms, at -0.32(meaning the relative response of OH is up to 0.32 times the relative difference in CH4, but opposite in direction). Regarding the use of the infinitesimal notation, this is just a straightforward reformulation of equation 2. This is now made explicit in the text.
13. Line 275: Is  $OH + CH_4$  the dominant OH sink in the model? What is the change in the kinetics of the reaction for the temperature change applied to the model?

The reaction coefficient for OH + CH4 in the model is  $1.85 \cdot 10^{-12} \exp(-1690/T)$ . This makes it one of the most temperature-sensitive reactions in the NIWA-UKCA chemistry scheme. The sensitivity of the reaction coefficient, at 290 K, evaluates to approximately 2%/K of temperature difference. This combined with the important role of CH4 + OH as a HOx sink, means that indeed CH4 + OH plays a major role in explaining the relatively small impact of temperature changes on OH. This is now made explicit in the text.

14. Line 310: What is the significance of this equation? Can it be applied to other models? Can values for the parameters be tabulated for various altitudes (or can altitude dependent parameters be given?). How valid is the assumption that the OH response is linear to changes in the forcings? As stated, Figure 8c suggests this is not a valid assumption.

This equation represents a working assumption; this is now made explicit in the text. We agree that there are some non-linearities, and the assumption of linearity is not perfect, but linearity explains almost the whole pattern in Figure 8 a. Other models that focus on global and background chemistry might well behave in ways similar to our model, but we haven't tested this.

15. Line 385: Please give some examples (and references!) of underestimated CH4 lifetimes by NIWA-UKCA and comparisons with other accepted estimates. An expanded introduction will help with this.

In the REF-C1 simulation of NIWA-UKCA used here, the CH4 lifetime evaluates to 7.6 years. This compares to 9.2 years as the best estimate from SPARC (2013). We have added text (in section 3.5) quantifying the CH4 lifetime in NIWA-UKCA and giving this best estimate. Further citations are added in the conclusions section.
- 16. Minor comments: Line 11: 'Its impact...', please change this to 'The impact of O3...' for clarity.
  Done
- 17. *Line 32: 'in-situ' to 'in situ'*. Done, also elsewhere.

18. Line 60: Please spell out NIWA in full.

Done

19. Line 71/line 135: What determines the concentrations of these species in the model if there are no emissions? Are they constrained to observations? Set to zero?

NIWA-UKCA model data are used for these species. This is now made explicit in the text (also in section 2.3).

20. Page 101: 'Vertically integrated ozone produced here' – please reword, do you mean 'produced in this way'?

We have changed 'vertically integrated ozone produced here' for 'the vertical integration of the observed O3 profiles'.

21. Line 161: Please replace 'a' and 'b' with 'k' in keeping with convention, and label the different 'k' appropriately to distinguish between reactions (i.e.  $k_a$ ,  $k_b$  or  $k_1$ ,  $k_2$ ).

Done

22. Line 290: Space in '5K'.

Done
- 23. *Line 329: 'sky' to 'skies'.* Done
- 24. *Line 336: Please change the word 'combinedly'.* We replace this with 'in combination'.
- 25. *Line 373: Please change 'chemical equilibrium' to 'chemical steady state'.* Done
- 26. *Figure 1: Panel e, please remove the degree symbol.* Done
- 27. Figure 2: Panel f, presumably this should refer to panels 2b and 2d? Done
- 28. Figure 3: Please remove the titles to the plots and leave just the labels a, b and c. See comments above regarding the exponential relationships please give the parameters (and fit statistics) for the relationships described if these are important. If they are, why mention them?

Done. We now state the fitting parameters, but we don't regard the fit statistics as necessary in this context as these coefficients are not used any further in the following text.

29. Figure 4: The data shown in the plots are given as percentages in the discussion. Please see comments above regarding consistency.

Changes of species are generally now given as percentages relative to the reference, but sensitivity coefficients are given as fractions. We think this is now handled consistently.
30. Figure 5: Please clarify in the caption that panels e and f refer to plots a & b and c & d, respectively. The analysis  $d \ln(OH)/d \ln(H_2O)$  is not explicitly referred to in the text (likewise for Figure 6).

We have clarified the caption and have removed the formula.

- 31. Figure 6: Figure 6e in the caption is referred to as Figure 6d.Changed. We have reformulated the caption along the same lines as figure 5.
- 32. *Figure 8: Panel c, please explain the significance of the dashed and red lines.* The significance of the red solid and black dashed lines has been explained in the caption of Figure 8.
Atmos. Chem. Phys. Discuss., doi:10.5194/acp-2016-448-AC1, 2016 © Author(s) 2016. CC-BY 3.0 License.

ACPD
General Comments: This paper shows the influence of biases in modeled  $O_3$ ,  $H_2O$ , CO,  $CH_4$ , and temperature on modeled OH as investigated using a single-column model and observations over Lauder, New Zealand. Model fields of the parameters listed above are replaced with observations, and the photochemical single-column model is used to re-calculate OH and establish changes and sensitivities in OH relative to a reference run. Impacts of O3 and temperature biases are further examined by separating kinetic and photolytic effects. Long-term OH trends and effects of clouds on OH are

**briefly examined.**

While this analysis is somewhat limited in scope and some aspects of the discussion are quite cursory, tropospheric OH is an important issue requiring varied and novel approaches to build on the community's understanding. With some revisions, this paper would contribute a useful method to help identify how model representation of OH can be improved and why model versus empirical estimates of the  $CH_4$  lifetime differ.

We agree with the reviewer that novel approaches to understanding the diversity of OH in global models are needed; we think the approach presented here is such a novel approach. We also accept that the scope is limited, namely we only assessed OH at the Lauder observatory. Within the context of this paper this is not easily addressed. However, in a follow-up study, the approach could be rolled put to other locations albeit with caution due to issues with data availability, a necessary reliance on satellite remote sensing data, or the role of other pollutants not present above Lauder. This is dwelt on a little more in the final section of the text now. We hope the revised paper now addresses the reviewer's concern.

Major comments:

1. Line 10: Please provide some quantification for these results. Particularly useful would be an indication of how much H2O differed between the model and observations as well as a quantification of how OH changed in response. The same could be done for subsequent species.

A quantification of biases in the key forcings and also of the OH sensitivity have been included in the abstract.

2. Line 47: You state two paragraphs above that "in-situ measurements of OH do not sufficiently constrain its global abundance." Here, you cite two Emmerson papers that do exactly that as justification for your SCM approach. I understand that constraining OH globally is not your aim, but the two statements still seem ACPD
contradictory. It would be worthwhile to strengthen your justification for this analysis - what questions are you seeking to answer? What role can this approach play in constraining global OH, even if there are limitations?

The Emmerson et al. papers are about using a box model to understand measurements of OH and  $HO_2$  taken in polluted environments. It would be incorrect to assert that these papers aim to quantify global OH. Our approach is unique in that we focus on the remote atmosphere and use long-term observations to constrain the model – other approaches such as Emmerson et al. have used campaign data. We have modified the text to this effect.

3. Line 75: The number of species and reactions represented in the NIWA-UKCA chemical mechanism seems low, at least compared to explicit schemes like the MCM (easily into the hundreds of species and thousands of reactions). Might be worth noting why it's important to maintain consistency with the NIWA-UKCA model/why you wouldn't want a more detailed mechanism in your SCM, since "assessing fast photochemistry" is your goal.

The purpose of this work is to assess the contribution of NIWA-UKCA biases to OH. If we used a different chemical mechanism to the one used in the NIWA-UKCA model, we would be introducing more uncertainties in the analysis. Also this might make the analysis less relevant to chemistry-climate modelling in general. A sentence reporting on this has been included in the text. See paragraph 1 of section 2.1.

4. Line 326: This section should be expanded. Even though the trends are not significant, they can still be quantified, and numbers here compared to values in the other studies you cite. Also, trends shown, for example, in Montzka et al., 2011 are derived from a globally, vertically integrated [OH] calculation, so separation into altitude bins, while useful, may not be the best comparison. I realize you don't seek to look at global [OH], but at least for this location, you could

**ACPD**
**include a vertically integrated OH trend to compare to Montzka et al. In addition to quantifying the trend, you could also quantify the interannual variability.**

We now state numerical values for the trends. We find no significant trend in the total column, which is in general agreement with a study on global OH by Montzka et al., however noting that we do not assess global OH. We now state the magnitude of interannual variability.

5. Line 430: The reader is likely interested to hear your hypotheses on why NIWA-UKCA is too moist and O3 is too high, even if further investigation is beyond the scope of this paper.

We appreciate that these are interesting though difficult questions. Difficulties with the hydrological cycle can be due to surface-atmosphere or cloudatmosphere interactions at Lauder. The ozone biases could be partly caused by the water vapour biases. We have added two sentences to this effect.

Minor comments:

6. Line 11: Reference to O3's kinetics and photolysis effects is unclear until defined in the body of the paper; please rephrase for abstract. Assertion that both are of similar magnitude does not seem well-supported, as pointed out above.

We have rephrased this in the abstract and now give numbers (in response to your earlier comment). These numbers do support the assertion that "both are of similar magnitude".

7. Line 12: Sentence about OH being inversely related to CO and CH4 is unnecessary for an audience familiar with OH.

We have removed this sentence.

8. Line 19: Use of "less-than-additive" is vague, especially for an abstract. Instead of focusing on how the LWC and IC effects combine, it would probably be more

**ACPD**
informative to note the quantitative results of the combined LIC simulation, if that is the more realistic one. This would likely be of greater interest to the reader. We have removed this sentence and now quantify the impact of clouds.

- 9. *Line 20: Please quantify trends as well, even if they are insignificant.* Done
- 10. Line 101: use of word "produced" is unclear.

The phrase has been replaced by "total column ozone calculated by integrating the observed  $O_3$  profiles".

11. Line 115: You make the case for not trusting radiosonde  $H_2O$  data above 8 km, but how about the NIWA-UKCA output? Does modeled  $H_2O$  agree well with FPH? A figure addressing this point might be suited for supplemental material.

NIWA-UKCA  $H_2O$  is subject to biases also above 8 km. We simply do not have any observational data above this level that is of high enough quality and frequency and covers the study period. We therefore do not discuss impacts on OH above 8 km here. This is now made explicit in the text.

12. Line 131: It would be helpful to address some anomalous behaviour in the  $H_2O$  profiles shown in Fig. 1: in the winter (presumably) of 1996, and to a lesser extent in other years, there are sudden high temperatures around 40-60 km – what's the cause of this? Is there evidence of this truly happening in the atmosphere or is it a result of interpolation, instrument artifact, etc?

These warming events may be the result of planetary wave breaking in the upper stratosphere. The event in 1996 is covered by many satellite measurements and therefore is likely real. We have not studied these events as they are outside the scope of this paper. We have added a sentence to the text to this effect.

ACPD
13. Line 137: What are the native temporal and spatial resolutions of this simulation? Do you also interpolate spatially?

We use 10-daily instantaneous output fields by NIWA-UKCA to construct the interpolates. The native resolution of NIWA-UKCA is  $3.75^{\circ} \times 2.5^{\circ}$ . We do not interpolate spatially. We consider the closest grid point to Lauder position. We have modified the text (section 2.1) to this effect.

- 14. *Line 174: Make clear that you're discussing local O*3, *or the "kinetics" effect.* We have modified the text to this effect.
- 15. Line 177: The sentence "The largest impact is in the free troposphere where these differences vary with altitude." is a bit vague. Please be specific; what are the differences you're referring to, and how do they vary?

We have removed this sentence and added more detail to this discussion.

16. Line 188: I'm not sure what you mean by the statement that kinetics and photolysis effects of the O3 bias are comparable. Based on my interpretation of the contours in Fig. 2, the response of OH to kinetic effects is both positive and negative, depending on the month and ranges from -12 to +4%; the response to photolysis effects is only positive, about 4 - -16%. The two effects somewhat cancel around Feb-June - is this what you're referring to? Please clarify.

We have qualified this statement. The magnitudes of the two effects are similar but seasonalities and height dependencies differ.

17. Line 241: The statistic that sensitivity of OH to CH4 changes peaks at  $\sim 40\%$  can be easily misinterpreted as the OH response; it may be helpful to highlight both the max OH response as well as the OH sensitivity to avoid confusion.

We now consistently express relative changes of OH,  $O_3$  etc. as percentages, and sensitivities such as this as fractions. We have added in the text that the maximum OH difference due to correcting CH4 is 0.6%.
18. Line 295: You stated above that the O'Connor et al. result may be due to cancellation of positive and negative temperature biases, but you show that temperatures at Lauder are cold-biased, so saying that your result of small impact of temp on OH corroborates that of O'Connor et al. seems like an apples-to-oranges comparison. I'd suggest reframing the discussion of O'Connor et al. – the small impact on OH in O'Connor et al. could have been due either to cancellation of temp biases or to low sensitivity of OH to temp changes, and your result suggests the latter? Or something to that effect.

We have followed the reviewer's suggestion.

19. Line 320: Care to hypothesize about what might be causing these nonlinearities?

Quite likely there are some feedbacks between correcting the water vapour and ozone biases which we now allude to in the text.

20. Line 366: what do you mean by "slow chemistry"? My best guess is something like oxidation of CH4 (long-lived), yet that is considered here, so I'm not sure about your intended meaning.

We supply independent initial states at every iteration of the SCM, meaning that species with a local chemical lifetime longer than the timestep of 1 hour can be considered prescribed for this purpose. Hence the model is only really suitable for fast radical chemistry. We have modified the text to this effect.

21. Line 382: Again, would like to see quantification here; how much is the H2O overestimated?

We now give this information in the text.

22. Line 387: Please include some references, particularly when citing "accepted literature estimates".

ACPD
Done

23. Line 400: "...small reduction...due to the strong dependence of OH + CH4 on temperature." This does not logically follow; you'd think, with a strong dependence, that you should see a large reduction. Please clarify.

It's been rephrased in paragraph 6 of the conclusions.

- 24. Line 406: Thank you for quantifying the H2O bias! I think this statistic would be better suited to earlier paragraph on H2O, plus repeat in Section 3 and in abstract. We have now added this information in the abstract and in section 3.
- 25. Fig. 2: The use of both blue and red for strictly positive values is slightly confounding at first glance (panels (c)-(f)); if possible, would help to keep the white contour at value 0 (applies to various upcoming figures as well).

We have changed all such plots displaying the OH sensitivity to changes in the forcings, i.e. keeping the white contour at value 0, and blue and red colour levels for negative and positive values respectively.

26. Table 1: Is the  $O_3$  photolysis effect analysis done in an altitude-dependent manner? I.e., is a new  $j_{O1D}$  value calculated at each vertical point based on an overhead  $O_3$  column that's adjusted to account for the strat column plus the partial tropospheric column overhead? I did not see any details regarding this in the text.

The  $j_{O1D}$  values are calculated at each altitude level, so it is taking into account the stratospheric contribution and the corresponding partial tropospheric column for each level. A sentence stating this has been added in the text. See section 2.1.

27. Fig. 5: Use of d  $\ln(OH)/d \ln(H2O)$  is not mentioned in text, is not consistent with " $A_i$ " terms in Fig. 4; please either justify switching metrics or maintain consistency
(same with Fig. 6).

We have added eq. 3, which basically states an alternative, equivalent way of formulating  $A_i$ .

Also we now consistently use percentages to indicate model biases (except for temperature) and fractions to indicate model sensitivities  $A_i$ .

28. Fig. 9: y-axis label is misleading since, based on the caption, this shows OH anomalies. Perhaps include word "Anomaly" or a "delta" sign. Also, the values chosen for the y-axis tick marks are not easy to work with; it would be nice if they were adjusted to lie along round calculation-friendly values (e.g. increments of 0.5 instead of 0.417 in panel c). Also, how are these anomalies calculated, relative to what?

We have relabelled plot 9c. These are absolute differences in units of  $10^5\,\rm molecules/cm^3.$

Technical corrections:

- 29. *Line 23: Use of word atmospher-e/-ic 3x* We have rephrased the sentence.
- 30. *Line 29: "plays a important" should be "plays an important"* Done
- 31. Line 50: "long time series" wording seems off; perhaps "long record of observation" instead. I'm also curious at this point, how long is long? Perhaps give an earliest year of observation.

This has been changed to "long records of measurements are available since 1986". We use data since 1994 in this work as MOPI1 measurements started in 1994.

ACPD
- 32. *Line 53: I think "Section 1" should be "Section 2"?* Corrected.
- 33. *Line 250: "altitide" should be "altitude"* Done
- 34. *Line 256: spelling of the word "assess" is incorrect* Done
- 35. Line 275: use of "explicitly" does not add meaning to this sentence but makes it read awkwardly; I suggest removing.

Done

- 36. *Line 279: "nearly completely linearly" should be "nearly linearly"* Done
- 37. *Line 324: instead of "altitude bands", I more often see the phrase "altitude bins"* Done
- 38. Line 378: use of word "effect" doesn't seem quite right; it's the bias you're correcting.

Done
Manuscript prepared for Atmos. Chem. Phys. with version 2015/04/24 7.83 Copernicus papers of the LATEX class copernicus.cls. Date: 7 October 2016

**Assessing the sensitivity of the hydroxyl radical to model biases in composition and temperature using a single-column photochemical model for Lauder, New Zealand**

L. López-Comí1,2, O. Morgenstern1,\*, G. Zeng1,\*, S. L. Masters2, R. R. Querel1, and G. E. Nedoluha3

1National Institute of Water and Atmospheric Research (NIWA), Lauder, New Zealand 2Department of Chemistry, University of Canterbury, Christchurch, New Zealand 3United States Naval Research Laboratory, Washington, DC, United States \*now at NIWA, Wellington, New Zealand

Correspondence to: O. Morgenstern (olaf.morgenstern@niwa.co.nz)

Abstract. We assess the major factors contributing to local biases in the hydroxyl radical (OH) as simulated by a global chemistry-climate model, using a single-column photochemical model (SCM) analysis. The SCM has been constructed to represent atmospheric chemistry at Lauder, New Zealand, which is representative of the background atmosphere of the Southern Hemisphere (SH)

- 5 mid-latitudes. We use long-term observations of variables essential to tropospheric OH chemistry, i.e. ozone (O3), water vapour (H2O), methane (CH4), carbon monoxide (CO), and temperature, and assess how using these measurements affect OH calculated in the SCM, relative to a reference simulation only using modelled fields. The analysis spans 1994 to 2010. Results show that OH responds approximately linearly to correcting biases in O3, H2O, CO, CH4, and temperature. The biggest im-
- 10 pact on OH is due to correcting an overestimation by approximately 20 to 60% of  $H_2O$ , using radiosonde observations. Correcting this moist bias leads to a reduction of OH by around 5 to 35%. This is followed by correcting predominantly overestimated  $O_3$ . In the troposphere, the model biases are mostly in the range of -10 to 30%. Its The impact of changing  $O_3$  on OH is due to two pathways; the OH responses to both are of similar magnitude but different
- 15 seasonality: Correcting in situ tropospheric ozone leads changes in OH in the range -14 to 4%, whereas correcting the photolysis rate of O3 in accordance with overhead column ozone changes leads to increases of OH of 8-16%. The OH sensitivityies to correcting CH4 and CO biases is inversely related to the relative changes applied to these species; are both 
[revised manuscript text omitted]

$$P(\text{HO}_{x}) \approx \frac{2k_{1}j_{O^{1}D}[\text{O}_{3}][\text{H}_{2}\text{O}]}{k_{2}[\text{O}_{2}] + k_{3}[\text{N}_{2}] + k_{1}[\text{H}_{2}\text{O}]}$$
(1)

- where k1 is the rate coefficient for O(1D) + H2O, jO1D is the rate of O3 photolysis producing O(1D), and k2 and k3 are the rate coefficients of quenching of O(1D) with O2 and N2, respectively (Liu and Trainer, 1988; Thompson et al., 1989; Madronich and Granier, 1992; Fuglestvedt et al., 1994). Accordingly, P(HOx) is affected by ozone changes principally in two different ways: Either locally through a change in [O3] or non-locally through a change in jO1D caused by changes in the
- 265 overhead total-column ozone (TCO). To separate the effects, we conduct three simulations with the SCM: The first simulation targets the local kinetics effect by applying changes in  $O_3$  concentrations but keeping all photolysis rates unchanged versus the reference simulation. A second simulation involves applying changes in  $j_{O^1D}$  according to changes in  $O_3$  (keeping the rest of photolysis rates unchanged), but considering a fixed  $O_3$  concentration, i.e. using the  $O_3$  concentrations of the refer-
- ence simulation. The  $j_{O^1D}$  calculation consistently takes into account absorption and scattering by stratospheric and tropospheric O3. A third simulation includes both effects simultaneously.

The results of these three sensitivity runs are displayed in Fig. 3. As expected, the pattern of  $O_3$  differences **between observed O3 and modelled O3** (Fig. 3a) is reflected in the pattern of

[revised manuscript text omitted]

$$A_i = \frac{\partial \ln[\mathbf{OH}]}{\partial \ln X_i}.$$
(3)

The sensitivity coefficients of OH to the kinetics and photolysis effects of  $O_3$  are shown in Fig. 5(a). Coefficient  $A_1$ , which represent the kinetics effect, ranges from 0 to 0.25 (meaning the relative response of OH is up to a quarter of the relative difference in  $O_3$ ). The sensitivity to photolysis  $(A''_1)$  is > 0.5 throughout much of the troposphere (meaning the relative response in OH is over half the relative change in  $j_{O^1D}$ ).

**330 3.2 OH sensitivity to H2O biases**

A perturbation simulation was performed using combined radiosonde and NIWA-UKCA  $H_2O$  (section 2.2). The OH response to correcting  $H_2O$  biases (Fig. 6) shows an approximately linear response with respect to the relative changes in  $H_2O$ , i.e. decreases in  $H_2O$  generally lead to a reduction of OH concentrations (eq. 1). Note that NIWA-UKCA substantially overestimates the radiosonde-observed

- 335 H2O vapour by up to 60% between 2 and 6 km (Fig. 6a); this translates into an overestimation of OH by up to  $\sim 40\%$  in the reference simulation (Fig. 6c) in the same region. The sensitivity of OH to changes in H2O (eq. 2) range from 5 to 0.5 in the troposphere (Figs. 6e and 5 (b) coefficient  $A_2$ ), with high sensitivity in the lower and free troposphere and reduced sensitivity in the tropopause region.
- 340 It is known that large uncertainties are associated with H2O vapour measurements. To illustrate this, we repeat the above simulation but now using European Centre for Medium–Range Weather Forecasts (ECMWF) ERA–Interim reanalysis (hereafter ERAI) (Dee and et al., 2011) H2O. Irrespectively of the large differences and the opposite signs in H2O biases between Lauder radiosonde

and ERAI data, the OH response to biases in H2O show approximately the same linear relationship

in both cases (Fig. 6). Likewise, the sensitivity of OH to changes in  $H_2O$  using ERAI data (Figs. 6f) and 5b, coefficient  $A_3$ ) resembles the sensitivity simulation using radiosonde  $H_2O$ .

**3.3 OH sensitivity to CH4 and CO biases**

The effect of CH4 changes on OH is displayed in Fig. 7 (a,c,e). The CH4 biases are generally small, up to only ~ 2%, and are assumed to be vertically uniform, with some seasonal variations. Decreases
in CH4 lead to increases in OH due to reduced loss of HOx by CH4 + OH. The response of OH to CH4 changes maximizes at 0.6% around 2 km, and decreases at higher altitudes. The seasonal variation of the OH response to CH4 biases maximizes in March/April (Fig. 7c), which coincides with the maximum absolute bias in CH4 (Fig. 7a) in the same months. However, the sensitivity of OH to CH4 changes maximizes in May/June (Fig. 7c), with a peak value of ~ 0.4. 
[revised manuscript text omitted]

Eq. (4) describes the relationship between the single-perturbation experiments and the all-forcings simulation, assuming that the OH response is linear to changes in the forcings. Figure 9 **a**,**b** indicates that the model responds approximately linearly to the combinations of all forcings, with OH

responses in the all-forcings simulation correlating at 0.9 with the sum of the OH responses in the individual-forcing simulations. Fig. 9(c) however also suggests that there are some notable non-linearity in the system chemistry of the troposphere at Lauder. Chemical feedbacks between the impacts of correcting water vapour and ozone may contribute to this non-linearity; for example, a change in the water vapour abundance may impact the sensitivity of OH to changing 445 O3.

430

**3.6 Trends in OH**

We examine variability and trends in OH using the SCM simulation including all key forcings separately for different altitude bands bins. The results (Fig. 10) indicate that there are no significant **long-term** trends in OH throughout the troposphere for the period of the simulation (1994-2010)

450 We find trends of  $-2.1 \pm 4.8\%$  at 0-2.5 km,  $0.9 \pm 2.3\%$  at 2.5-5 km,  $2.6 \pm 3.5\%$  at 5-7.5 km, and  $3.6 \pm 4.1\%$  at 7.5-10 km over the period of 1994-2010), but there is evidence substantial interannual variability of interannual variations at all altitudes. This is in agreement with other studies (
[revised manuscript text omitted]
 + CH4 on temperature (eq. 1). However, for NIWA-UKCA the impact of temperature biases impact on OH at Lauder is small, the impact of this reaction on OH is buffered by other less temperature-dependent reactions, causing only a small sensitivity of OH to temperature. This is in agreement with O'Connor et al. (2009).

- The results of the simulation considering simultaneous changes in all the key forcings indicate that 540 OH responds approximately linearly to all the major forcings that contribute to the oxidising capacity of the atmosphere. We find that biases in  $O_3$ ,  $H_2O$ ,  $CH_4$ , CO, and temperature all affect the oxidising capacity of the atmosphere at Lauder, with  $H_2O$  and  $O_3$  biases dominating. The NIWA-UKCA model generally produces a moist bias (by  $\sim 0 - 50\%$ ) relative to radiosonde measurements; this leads to an overestimation of OH of up to 40%. This makes water vapour a leading contender
- to explain the underestimated global lifetime of  $CH_4$  in NIWA-UKCA (Morgenstern et al., 2013; Telford et al., 2013). 
[revised manuscript text omitted]